# Application of ATAC-Seq for genome-wide analysis of the chromatin state at single myofiber resolution

Korin Sahinyan[1,2†], Darren M Blackburn[1,2†], Marie-Michelle Simon[1,3†], Felicia Lazure[1,2], Tony Kwan[1,3], Guillaume Bourque[1,3,4], Vahab D Soleimani[1,2]*

[1]Department of Human Genetics, McGill University, Montreal, Canada; [2]Lady Davis Institute for Medical Research, Jewish General Hospital, Montreal, Canada; [3]McGill Genome Centre, Montreal, Canada; [4]Canadian Centre for Computational Genomics, Montreal, Canada

**Abstract** Myofibers are the main components of skeletal muscle, which is the largest tissue in the body. Myofibers are highly adaptive and can be altered under different biological and disease conditions. Therefore, transcriptional and epigenetic studies on myofibers are crucial to discover how chromatin alterations occur in the skeletal muscle under different conditions. However, due to the heterogenous nature of skeletal muscle, studying myofibers in isolation proves to be a challenging task. Single-cell sequencing has permitted the study of the epigenome of isolated myonuclei. While this provides sequencing with high dimensionality, the sequencing depth is lacking, which makes comparisons between different biological conditions difficult. Here, we report the first implementation of single myofiber ATAC-Seq, which allows for the sequencing of an individual myofiber at a depth sufficient for peak calling and for comparative analysis of chromatin accessibility under various physiological and disease conditions. Application of this technique revealed significant differences in chromatin accessibility between resting and regenerating myofibers, as well as between myofibers from a mouse model of Duchenne Muscular Dystrophy (mdx) and wild-type (WT) counterparts. This technique can lead to a wide application in the identification of chromatin regulatory elements and epigenetic mechanisms in muscle fibers during development and in muscle-wasting diseases.

## Editor's evaluation

The authors have described an innovative application of ATAC-Seq for genome-wide analysis of the chromatin state at single myofiber resolution.

*For correspondence:
vahab.soleimani@mcgill.ca

†These authors contributed equally to this work

Competing interest: The authors declare that no competing interests exist.

## Introduction

Skeletal muscle evolved for contraction and the production of force. The main component of skeletal muscle are myofibers which are formed from the fusion of myogenic precursor cells (*Buckingham et al., 2003*) resulting in large postmitotic syncytia that are composed of repeating contractile units, called sarcomeres (*Huxley and Hanson, 1954*). Myofibers exhibit wide variations in their metabolic activity and contractile properties (*Zierath and Hawley, 2004*). In addition, they have a highly adaptive nature where their size, myosin heavy chain isoform, energy metabolism, and the overall skeletal muscle mass, among other characteristics, are regulated by complex processes involving rates of protein turnover (*Bohé et al., 2001*; *Mittendorfer et al., 2005*), as well as transcriptional (*Quiat et al., 2011*) and posttranscriptional (*Weskamp et al., 2021*) control of gene expression. Due to their adaptive nature, myofibers can change in response to exercise (*Zierath and Hawley, 2004*;

*Wilson et al., 2012*; *Dons et al., 1979*), aging (*Deschenes, 2004*) and diseases, such as sarcopenia (*Thompson, 2002*; *Nilwik et al., 2013*) and cachexia (*Roberts et al., 2013*). Therefore, the study of the myofiber transcriptome and epigenome can provide key insights into how skeletal muscle adapts and changes under various stimuli, and it can potentially lead to the discovery of novel therapeutic venues for muscle related diseases.

Myofibers also act as a key signaling component of muscle stem cells (MuSCs) (*Lazure et al., 2020*), which are in turn required for the regeneration of muscle fibers after injury (*Snow, 1977*; *Shadrach and Wagers, 2011*; *Dumont et al., 2015*). Skeletal muscle is a very heterogenous tissue composed not only of myofibers and their associated MuSCs, but also numerous different non-myogenic cell types (*Giordani et al., 2019*). Previous studies using whole muscle sequencing captures not only the myofibers but also the other resident cell types in the muscle, making it challenging to attribute any changes in the transcriptome and epigenome specifically to myofibers as they could be due to changes in these other cell types. Recent advances in Next Generation Sequencing (NGS) now allow for high-dimensional analysis at a single-cell level. Recent studies using these technologies to study muscle tissue, such as single nucleus RNA-Seq and single nucleus ATAC-Seq have analyzed the transcriptome and epigenome of the myonuclei within the muscle fiber (*Petrany et al., 2020*; *Dos Santos et al., 2020*; *Kim et al., 2020*). However, they present certain limitations where they sequence all myonuclei present in the muscle and cannot distinguish between different myofibers, as well as having low sequencing depth with a limited capacity for downstream analyses.

The chromatin state plays a key role in transcriptional regulation and the determination of cellular identity (*Zhu et al., 2018*). Although the accessible regions make up only 3% of the total genome, it represents over 90% of known transcription factor binding sites (*Thurman et al., 2012*). Chromatin accessibility is a determinant of gene expression, and changes in chromatin accessibility have been identified in different biological and disease conditions such as during development (*Liu et al., 2019a*; *Trevino et al., 2020*), cancers (*Corces et al., 2018*; *Rendeiro et al., 2016*), and neurological disorders (*Wang et al., 2020*; *Bastle and Maze, 2019*). Thus, in recent years, the study of epigenetics and chromatin accessibility has become a promising field for the development of novel therapeutics. Today, ATAC-Seq is a widely used method that allows for the mapping of the accessible chromatin regions in the genome. ATAC-Seq relies on the hyperactive Tn5 transposase that fragments the accessible regions in the genome while simultaneously ligating sequencing compatible adaptors (*Corces et al., 2017*; *Buenrostro et al., 2015*). Over the years, ATAC-Seq has been applied to many different cell types and tissues (*Liu et al., 2019b*; *Yan et al., 2020*; *Rocks et al., 2022*). However, to our knowledge, it has not been performed on a single myofiber, possibly due to the rigidity of their membrane, high levels of mitochondria (*Janssen et al., 2000*; *Ortenblad et al., 2018*; *Mishra et al., 2015*), and the low number of myonuclei that are present in a single myofiber (*Neal et al., 2012*; *Cramer et al., 2020*).

Here, we have adapted OMNI-ATAC-seq to determine the genome-wide chromatin accessibility of myonuclei contained within a single Extensor Digitorum Longus (EDL) muscle fiber of a mouse. The single myofiber ATAC-Seq (smfATAC-Seq) method that we applied in this study allows for the investigation of the accessible chromatin regions of a single myofiber, without the presence of other confounding cell types. The smfATAC-Seq has a sequencing depth of approximately 6 million final reads aligned which provides approximately 30,000 peaks called. Using this method, we provide comparative analysis of chromatin accessibility between resting and regenerating myofibers, as well as their MuSC progenitors. In addition, application of this method to the study of myofibers isolated from a mouse model of Duchenne Muscular Dystrophy (mdx) (*McGreevy et al., 2015*) and their WT counterparts provide a genome-wide assessment of changes in chromatin accessibility in Duchenne Muscular Dystrophy (DMD). This method can be used in the future to profile the epigenetic state of myofibers in different disease conditions, and under various physiological and physical stimuli, and to identify active cis-regulatory elements in muscle fibers.

## Results
### Generation of ATAC-Seq libraries from a single myofiber
A single EDL myofiber of a mouse contains an average of 200–300 myonuclei (*Neal et al., 2012*; *Cramer et al., 2020*), making genome-wide analyses of the chromatin state difficult. With the

advancements in next generation sequencing (NGS) and the development of the OMNI ATAC-Seq protocol (*Corces et al., 2017*), analysis of chromatin accessibility of samples with an input of as low as 500 cells is now possible (*Corces et al., 2017*). However, myofibers present additional challenges with their rigid membrane and high levels of mitochondria (*Janssen et al., 2000*; *Ortenblad et al., 2018*; *Mishra et al., 2015*). Here, we report a robust protocol for the successful application of ATAC-Seq on a single myofiber isolated from the EDL muscle. Our method relies on the lysis and permeabilization of a single myofiber followed by transposition with a hyperactive Tn5 transposase (*Corces et al., 2017*) (*Figure 1*). DNA fragment sizes obtained from the smfATAC-Seq were of a similar range in size as those obtained from conventional OMNI ATAC-Seq which we have performed on 5000 MuSCs that were freshly isolated by Fluorescence Activated Cell Sorting (FACS) (*Figure 1—figure supplement 1*). Furthermore, analysis showed that only 0.9–2.09% of reads were derived from the mitochondria in smfATAC-Seq (*Table 1*), suggesting that this method is highly efficient for the removal of mitochondria from mitochondria-rich myofibers. Following the removal of mitochondrial reads, there were approximately 6 million final reads aligned and 30,000 peaks called, demonstrating a sufficient sequencing depth for downstream analysis (*Table 1*).

## smfATAC-Seq can be used to study chromatin accessibility of myofibers under different physiological conditions

In addition to adapting the OMNI ATAC-Seq method to study the chromatin accessibility of a single myofiber, we also demonstrate the application of this technique for comparative analysis of chromatin accessibility between myofibers under different conditions. For that purpose, we performed ATAC-Seq on myofibers that were in a resting (uninjured) or regenerating (injured) state. Uninjured and injured (7 days post cardiotoxin (CTX) induced injury) myofibers were isolated from wild-type C57BL/6 mice and smfATAC-Seq was performed to compare the changes in chromatin accessibility during regeneration. In addition, as a further quality control, we compared the chromatin accessibility between myonuclei within a single myofiber and 5000 freshly isolated MuSCs. This analysis not only identified accessible regions of chromatin in myofibers and MuSCs, but it also revealed a repertoire of active cis-regulatory elements in each sample.

Apart from the myofibers and their associated MuSCs, skeletal muscle also contains many non-myogenic cells such as endothelial cells, adipocytes, hematopoietic cells, fibroblasts, fibro/adipogenic progenitors (FAPs), and macrophages (*Giordani et al., 2019*; *De Micheli et al., 2020a*; *De Micheli et al., 2020b*). Our smfATAC-Seq method allows for the analysis of chromatin accessibility of a single myofiber without the confounding effect of these contaminating cell types. Given that the whole muscle contains non-myogenic cell types, we first compared smfATAC-Seq to an ATAC-Seq performed on whole EDL muscle by *Ramachandran et al., 2019* (GSM3981673) (*Ramachandran et al., 2019*) for the enrichment of ATAC-Seq peaks on the genes of non-myogenic cells. We obtained the list of genes that are solely expressed in the whole muscle (RPM of at least 10) but not in the myofibers (RPM of 0) by using an RNA-Seq dataset performed on whole muscle and a single myofiber by *Blackburn et al., 2019* (GSE138591) (*Blackburn et al., 2019*). This list represents genes that are only expressed by the muscle resident non-myogenic cell types, designated as "non-fiber muscle genes". We determined the number of peaks in smfATAC-Seq that overlap with non-fiber muscle genes, which revealed that only 0.1% of the peaks overlapped with the top 100 non-fiber muscle genes (*Table 2*). In comparison, 0.33% of the peaks in the whole EDL muscle ATAC-Seq (GSM3981673) (*Ramachandran et al., 2019*) overlapped with the top 100 non-fiber muscle genes (*Table 2*). The significant difference in the overlap with the non-fiber muscle genes between the whole muscle ATAC-Seq and smfATAC-Seq suggest that the whole muscle ATAC-Seq has enrichment of peaks associated with non-myogenic genes when compared to the smfATAC-Seq, which implies that smfATAC-Seq can successfully exclude the non-myogenic cell types. In contrast, the number of overlapping peaks with all the genes expressed in whole muscle in the EDL ATAC-Seq and smfATAC-Seq were similar (*Table 2*). To further illustrate the absence of the non-myogenic cell types in the smfATAC-Seq samples, peaks at the promoter regions of marker genes of muscle resident cells were searched for. Specifically, Platelet and Endothelial Cell Adhesion Molecule 1 (*Pecam1*) was used to determine whether endothelial cells were present (*Khan et al., 2005*). Similarly, Resistin (*Retn*) and *Cd45* were used as markers for adipocytes and hematopoietic cells, respectively (*Steppan et al., 2001*; *McKinney-Freeman et al., 2009*). The cell Surface Antigen *Thy1* was the marker selected for fibroblasts (*Agorku et al., 2019*). In addition,

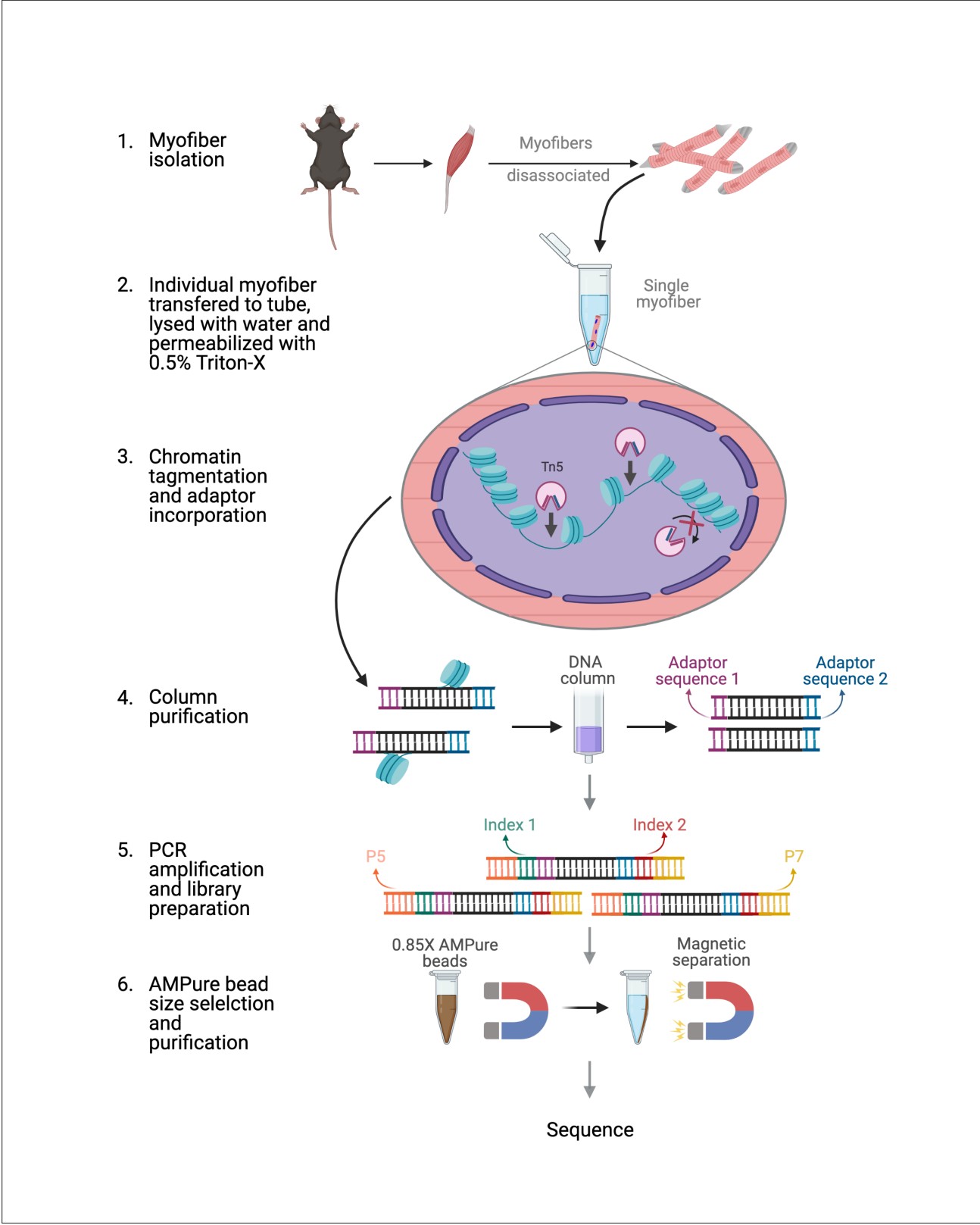

1. Myofiber isolation

   Myofibers disassociated

   Single myofiber

2. Individual myofiber transferred to tube, lysed with water and permeabilized with 0.5% Triton-X

3. Chromatin tagmentation and adaptor incorporation

   Tn5

4. Column purification

   DNA column · Adaptor sequence 1 · Adaptor sequence 2

5. PCR amplification and library preparation

   Index 1 · Index 2 · P5 · P7

6. AMPure bead selelction and purification

   0.85X AMPure beads · Magnetic separation

   Sequence

**Figure 1.** Schematic of ATAC-seq performed on a single myofiber. Schematic of the steps and reactions involved in the preparation of sequencing ready libraries of single myofiber DNA for ATAC-Seq. Briefly, myofibers were isolated from the EDL muscle and an individual myofiber was transferred to a 0.2 mL microtube. The myofiber was then lysed with ddH2O and the myonuclei were permeabilized with 0.5% Triton X-100. Then, open chromatin regions were tagmented with hyperactive Tn5 transposase and the DNA fragments were purified through column purification. The tagmented DNA was

*Figure 1 continued on next page*

*Figure 1 continued*

then amplified by PCR and Nextera adaptors were incorporated. Finally, size selection and purification were performed using 0.85 X AMPure beads, resulting in sequencing ready libraries. Figure was made using BioRender.

The online version of this article includes the following figure supplement(s) for figure 1:

**Figure supplement 1.** Quality control of ATAC-Seq libraries.

Lymphocyte antigen 6A (*Ly6a*) and Adhesion G-protein-coupled receptor E1 (*Adgre1*) were selected for fibro/adipogenic progenitors (FAPs) and macrophages, respectively (*Waddell et al., 2018*; *Joe et al., 2010*). None of these marker genes had ATAC-Seq peaks at their promoters, indicating that only a single myofiber is processed without any other contaminating cell types (*Figure 2—figure supplement 1*).

smfATAC-Seq can be successfully applied to myofibers under different conditions. For instance, we applied smf-ATAC-Seq to analyze the chromatin accessibility of myofibers under resting and CTX-mediated injury conditions. In a disease condition or in injury, not all myofibers undergo damage or regenerate to the same degree. Therefore, individual myofibers within a muscle can be in different physiological and disease conditions asynchronously (*Folker and Baylies, 2013*). Damaged or regenerating myofibers can be visualized by their characteristic feature of centrally located nuclei (*Roman and Gomes, 2018*). Injured myofibers in this method were visually selected for the presence of the centrally located myonuclei by Hoechst staining and the selected myofiber was used for downstream processing with smfATAC-Seq (*Figure 2A and B*). The selection of a specific myofiber that our smfATAC-Seq allows for, as well as the application of trypsin to remove any associated cells that may be present, results in the sequencing of DNA fragments corresponding purely to the myonuclei within a specific myofiber.

## smfATAC-Seq can identify the accessible chromatin regions of a single myofiber

To validate the quality of the ATAC-Seq data generated from a single EDL myofiber, we first investigated the profiles of the ATAC-seq samples from both injured and uninjured myofibers as well as freshly sorted MuSCs. We investigated the similarity between biological replicates for each condition (i.e. uninjured and injured myofibers and MuSCs) by visualization of ATAC-Seq peaks for the muscle-specific gene muscle creatine kinase (*Ckm*) (*Tai et al., 2011*), the housekeeping gene *Gapdh* and the MuSC specific gene myogenic factor 5 (*Myf 5*) (*Rudnicki et al., 1993*) (*Figure 2—figure supplement 2A-C*). This analysis not only indicates that smfATAC-Seq can reliably detect chromatin accessibility in a single myofiber but also the presence of comparable peaks in each biological replicate within the specific condition shows the similarity between the samples (*Figure 2—figure supplement 2A-C*). In addition, Pearson correlation analysis between the biological replicates showed a high correlation of ATAC-seq reads between the replicates, indicating consistency within the samples (*Figure 2—figure supplement 2D-J*). Furthermore, we mapped ATAC-Seq peaks with DNase-Seq from skeletal muscle (Sequence Read Archive, accession # SRX191047) and the similarity between our ATAC-Seq and previous DNase-Seq was confirmed through the common peaks present for representative genes, as visualized on the IGV (*Figure 2—figure supplement 2A-C*). We also analyzed the overlap between the smfATAC-Seq on EDL myofibers with the ATAC-Seq performed on the whole EDL muscle by *Ramachandran et al., 2019* (GSM3981673) (*Ramachandran et al., 2019*). This analysis revealed that 65% of the smfATAC-Seq peaks in the uninjured myofibers overlap with the whole EDL muscle ATAC-Seq (*Table 3*).

Accessible chromatin regions are associated with various histone marks such as H3K27ac and H3K4me3 (*Zhang et al., 2015*; *Berger, 2007*; *Barrera et al., 2008*). Thus, we compared the smfATAC-Seq to publicly available datasets of ChIP-Seq on H3K27ac in EDL muscle that was previously performed by *Ramachandran et al., 2019* (GSM3515022, GSM3515023) (*Ramachandran et al., 2019*). The comparative analysis has revealed that there were only 97 peaks in the smfATAC-Seq that did not overlap with the H3K27ac peaks, while the majority of the peaks, 6090 peaks, were common to the H3K27ac peaks present in the entire EDL muscle (*Figure 2—figure supplement 2K*). This demonstrates that the accessible regions that are assessed by smfATAC-Seq correspond to the regions of the chromatin marked by histones that are associated with open chromatin such as H3K27ac. Overall,

**Table 1.** Sequencing read information for smfATAC-Seq and MuSCs ATAC-Seq libraries.

| Library | Number of raw reads | Number of surviving reads | Aligned filtered reads (mm10 reference) | Duplicate reads | Mitochondrial reads | Percentage of mitochondrial reads (%) | Final reads aligned | Number of peaks | Fraction in peaks (FrIP) |
|---|---|---|---|---|---|---|---|---|---|
| Muscle Stem Cells_1 | 175924734 | 113938436 | 103130186 | 47623836 | 529,967 | 0.51 | 54976383 | 65,568 | 0.3642 |
| Muscle Stem Cells_2 | 174965936 | 117357212 | 103570009 | 43672484 | 374,176 | 0.36 | 59523349 | 68,658 | 0.1971 |
| Muscle Stem Cells_3 | 131990380 | 91261584 | 79944121 | 31299456 | 223,540 | 0.28 | 48421125 | 69,573 | 0.1296 |
| Injured_1 | 229935426 | 117212678 | 90040002 | 81024926 | 830,215 | 0.92 | 8184861 | 32,853 | 0.2885 |
| Injured_2 | 194563870 | 129934972 | 98752157 | 88549329 | 1300615 | 1.32 | 8902213 | 28,351 | 0.2863 |
| Injured_3 | 142411536 | 62888552 | 52132455 | 42271079 | 868,808 | 1.67 | 8992568 | 25,002 | 0.2325 |
| Uninjured_1 | 145465410 | 75781456 | 61034569 | 52588315 | 1274332 | 2.09 | 7171922 | 12,276 | 0.2181 |
| Uninjured_2 | 151015852 | 64192706 | 50120282 | 45914841 | 965,037 | 1.93 | 3240404 | 14,742 | 0.3208 |
| MDX_1 | 107540762 | 50979732 | 40485803 | 36205908 | 802,561 | 1.98 | 3477334 | 40,833 | 0.7256 |
| MDX_2 | 103130726 | 54209722 | 46455472 | 37291531 | 1099747 | 2.37 | 8064194 | 39,254 | 0.4932 |
| MDX_3 | 108130662 | 48920904 | 40677359 | 34484003 | 1171316 | 2.88 | 5022040 | 35,691 | 0.5589 |
| WT_1 | 104219578 | 43914902 | 34162142 | 28600498 | 1651199 | 4.83 | 3910445 | 26,873 | 0.7283 |
| WT_2 | 110108692 | 37411936 | 31299222 | 25345321 | 1143317 | 3.65 | 4810584 | 28,430 | 0.64 |
| WT_3 | 183583506 | 72489354 | 56983637 | 49310923 | 1840265 | 3.23 | 5832449 | 39,178 | 0.7611 |
| WT_4 | 86533840 | 36708706 | 28714893 | 25157965 | 1404712 | 4.89 | 2152216 | 21,252 | 0.752 |

**Table 2.** Percentage of ATAC-Seq peaks that overlap with the TSS±500 bp by at least 1 bp.

| | Top 100 genes expressed in whole muscle but not in myofiber | | | Top 50 genes expressed in whole muscle but not in myofiber | | | All genes expressed in whole muscle tissue | | | All genes in the genome | | |
|---|---|---|---|---|---|---|---|---|---|---|---|---|
| | Number of overlapping peaks | Total number of peaks | % overlapping peaks | Number of overlapping peaks | Total number of peaks | % overlapping peaks | Number of overlapping peaks | Total number of peaks | % overlapping peaks | Number of overlapping peaks | Total number of peaks | % overlapping peaks |
| Uninjured_Fiber | 12 | 19,704 | 0.0609013 | 3 | 19,704 | 0.0152253 | 7,865 | 19,704 | 39.915753 | 12,995 | 19,704 | 65.951076 |
| Injured_Fiber | 65 | 47,112 | 0.1379691 | 12 | 47,112 | 0.0254712 | 14,259 | 47,112 | 30.266174 | 26,198 | 47,112 | 55.607913 |
| EDL_Whole_Muscle | 198 | 60,719 | 0.3260923 | 65 | 60,719 | 0.1070505 | 18,419 | 60,719 | 30.334821 | 33,048 | 60,719 | 54.427774 |

Genes identified as being expressed solely in whole muscle but not in myofiber were retrieved from "High-resolution genome-wide expression analysis of single myofibers using SMART-Seq, *JBC*, **Blackburn et al., 2019**" and were defined as any gene with an expression of at least 10 RPM in the whole muscle RNA-seq, but 0 RPM in the single myofiber RNA-seq. All genes expressed in whole muscle tissue was defined as any gene that had an RPM value of at least 10 RPM from the whole muscle RNA-seq data by **Blackburn et al., 2019** accessible through the GEO accession number GSE138591.

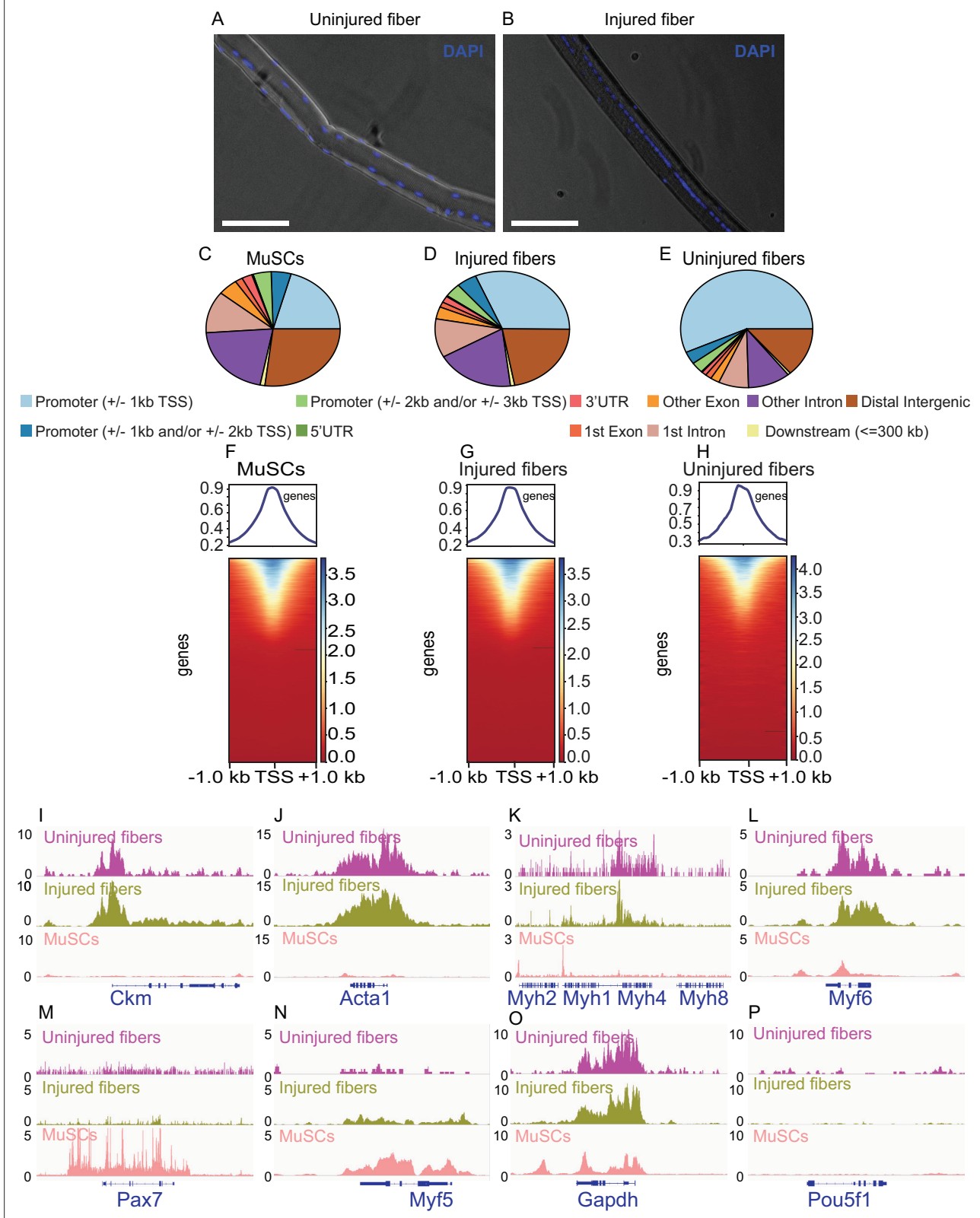

**Figure 2.** smfATAC-Seq can effectively identify the accessible regions on a single myofiber. (**A**) Representative picture of an isolated WT C57BL/6 J uninjured myofiber stained for Hoechst showing the presence and location of myonuclei. Scale bar = 50 μm. (**B**) Representative picture of an isolated WT C57BL/6 J injured myofiber (7 days post cardiotoxin induced injury) stained for Hoechst showing centrally located myonuclei as a marker of a regenerating fiber. Scale bar = 50 μm. Hoechst stain was visualized on the DAPI channel. (**C–E**) Peak annotation pie charts for ATAC-Seq peaks of

*Figure 2 continued on next page*

*Figure 2 continued*

MuSCs, injured myofibers and uninjured myofibers, respectively. (**F–H**) Heatmaps showing enrichment at transcription start site (TSS) for the ATAC-Seq libraries of MuSCs, injured myofibers and uninjured myofibers, respectively. (**I–P**) IGV snapshots of known genes expressed in muscle fiber and/or MuSCs displaying accessibility on their respective TSS. (**I**) The muscle creatine kinase (*Ckm*). (**J**) Actin alpha 1 (*Acta1*). (**K**) Part of the myosin heavy chain (*Myh*) gene cluster. (**L**) Myogenic factor 6 (*Myf6*). (**M**) Paired Box 7 (*Pax7*). (**N**) Myogenic factor 5 (*Myf5*). (**O**) Housekeeping gene *Gapdh*. (**P**) POU Class 5 homeobox 1 (*Pou5f1*) as a negative control. *ATAC-Seq was performed in biological replicates (n = 3 MuSCs, n = 3 injured myofibers, n = 2 uninjured myofibers).

The online version of this article includes the following figure supplement(s) for figure 2:

**Figure supplement 1.** IGV snapshots of non-myogenic genes.

**Figure supplement 2.** Correlation analysis between biological replicates of each condition.

**Figure supplement 3.** IGV snapshots of Myogenic Regulatory Factors (MRFs).

these analyses suggest that smfATAC-Seq can robustly measure chromatin accessibility and identify active cis-regulatory elements in a single EDL myofiber.

Following the initial quality control and the correlation analysis, the biological replicates from the same condition were pooled for further analysis. Peak annotation analysis for MuSCs revealed that more than half of the peaks were in the intron/distal intergenic regions (i.e. enhancer regions) and about 25% of the peaks were in the promoter region (*Figure 2C*, *Table 4*). Peak annotations for the uninjured and injured single myofibers also showed a great proportion of peaks in the enhancer and promoter regions (*Figure 2D and E*, *Table 4*). In addition, an enrichment of ATAC-seq reads around Transcription Start Sites (TSS) (±1 kb) from all datasets was observed, which is a typical result that is expected from ATAC-Seq (*Yan et al., 2020*) (*Figure 2F–H*).

To further assess the quality of the ATAC-Seq data, we analyzed select genes that are expressed by either MuSCs or myofibers. For instance, in the myofiber samples, we confirmed the presence of ATAC-seq peaks in the promoter regions of *Ckm*, Actin alpha 1 (*Acta1*), Myogenic factor 6 (*Myf6*), and Myosin heavy chain 4 (*Myh4*), all of which are expressed by myofibers but not MuSCs (*Lazure et al., 2020*; *Tai et al., 2011*; *Nowak et al., 1999*; *Stuart et al., 2016*) (*Figure 2I–L*). On the other hand, in MuSCs we observed peaks in the promoter regions of Paired Box 7 (*Pax7*), and *Myf5*, genes that are known to be expressed in MuSCs (*Rudnicki et al., 1993*; *Seale et al., 2000*) (*Figure 2M–N*). *Gapdh* was used a housekeeping gene for all samples (*Figure 2O*) and *Pou5f1*, a marker of pluripotency, was used as a negative control (*Figure 2P*). These observed peaks for known expressed genes demonstrate that our method, smf-ATAC-Seq, can reliably analyse chromatin accessibility in a single myofiber.

Muscle regeneration and repair rely on the temporal expression of Myogenic Regulatory Factors (MRFs), *Myf5*, *MyoD*, *Myog*, and *Myf6/MRF4* (*Hernández-Hernández et al., 2017*; *Montarras et al., 1991*; *Asfour et al., 2018*). Therefore, we assessed the chromatin accessibility of the MRFs in MuSCs

**Table 3.** Percentage of overlapping peaks between smfATAC-Seq from uninjured myofibers and whole EDL muscle ATAC-Seq.

| | Percent overlap (%) |
|---|---|
| smfATAC-Seq peaks that overlap with EDL-ATAC-Seq by at least 1 bp | 65.9510759 |
| smfATAC-Seq peaks that overlap with EDL-ATAC-Seq by at least 20% | 61.4951279 |
| smfATAC-Seq peaks that overlap with EDL-ATAC-Seq by at least 40% | 52.6136825 |
| smfATAC-Seq peaks that overlap with EDL-ATAC-Seq by at least 60% | 42.1082014 |
| smfATAC-Seq peaks that overlap with EDL-ATAC-Seq by at least 90% | 24.3453106 |

Whole EDL muscle ATAC-Seq was retrieved from "Dynamic enhancers control skeletal muscle identity and reprogramming, *Ramachandran et al., 2019*." This data is accessible through the GEO accession number GSM3981673.

**Table 4.** Percentage of total peaks found in each genomic feature.

| | Muscle stem cells (%) | Injured myofiber (%) | Uninjured myofiber (%) | MDX myofiber (%) | WT myofiber (%) |
|---|---|---|---|---|---|
| Promoter (±1 kb TSS) | 20.66 | 31.61 | 56.54 | 35.58 | 35.15 |
| Promoter (±1 kb and/or ±2 kb TSS) | 4.81 | 4.84 | 3.45 | 3.78 | 4.53 |
| Promoter ((±2 kb and/or ±3 kb TSS)) | 4.37 | 3.92 | 3.01 | 4.14 | 4.30 |
| 5'UTR | 0.34 | 0.27 | 0.23 | 0.46 | 0.39 |
| 3'UTR | 2.50 | 1.82 | 1.15 | 2.86 | 2.58 |
| First Exon | 1.83 | 1.47 | 1.53 | 1.94 | 1.78 |
| Other Exon | 4.75 | 3.42 | 2.19 | 4.74 | 4.25 |
| First Intron | 11.85 | 10.87 | 7.35 | 10.93 | 10.56 |
| Other Intron | 20.80 | 18.84 | 10.35 | 18.81 | 18.30 |
| Downstream ( ≤ 300 kb) | 1.16 | 1.01 | 0.69 | 1.02 | 0.99 |
| Distal Intergenic | 26.95 | 21.93 | 13.51 | 15.74 | 17.15 |

and in the myofibers under homeostasis and regeneration (*Figure 2—figure supplement 3*). We observed peaks in the promoter regions of *Myf5* only in the MuSCs but not in the myofibers and peaks in the promoters of *Myog* and *Myf6/MRF4* were solely observed in the myofibers (*Figure 2—figure supplement 3*). However, we observed peaks in the promoter regions of *MyoD* in both the MuSCs and myofibers (*Figure 2—figure supplement 3*).

## Uninjured and injured myofibers and MuSCs display distinct chromatin states

To show the global differences in chromatin accessibility between MuSCs, uninjured and injured myofibers, we first performed heatmap clustering of Pearson correlation coefficients on all the replicates/samples, which shows that the biological replicates within conditions are more similar to one another than to those from the other conditions (*Figure 3A*). This can also be observed through Principal Component Analysis (PCA) where each condition clusters separately, with the injured and uninjured myofibers being more similar to one another than to MuSCs (*Figure 3B*). To test whether the differences between regenerating and resting myofibers are overshadowed by their differences with MuSCs, we performed heatmap clustering of Pearson correlation coefficients and PCA analysis for injured and uninjured myofibers only, without MuSCs (*Figure 3—figure supplement 1*). This further highlighted how the uninjured and injured myofibers cluster separately (*Figure 3—figure supplement 1*).

To ensure that the differences seen between myofibers were not due to differences in fiber types, we investigated the chromatin accessibility of known marker genes for slow and fast fiber types. Troponin I2 (*Tnni2*) and Troponin T3 (*Tnnt3*), markers of fast fiber types, (*Mullen and Barton, 2000*; *Wei and Jin, 2016*) had a high level of chromatin accessibility while Troponin T1 (*Tnnt1*) and Myosin heavy chain 7 (*Myh7*), which are expressed in slow fiber types displayed no chromatin accessibility (*Wei and Jin, 2016*; *Meredith et al., 2004*) (*Figure 3—figure supplement 2*). This data indicates that only fast fiber types were analyzed in this study and that the differences in the chromatin state between the injured and uninjured myofibers were not due to the differences in the fiber types.

Differential analysis was performed on the ATAC-Seq peaks based on the regions defined by the consensus peak sets derived from the uninjured and injured myofibers and MuSCs conditions. The clustering analysis based on the consensus peak set shows the overall chromatin state differences between the MuSCs and injured and unjured myofibers where the replicates within a condition are more similar to one another than to those of the other conditions (*Figure 3C*).

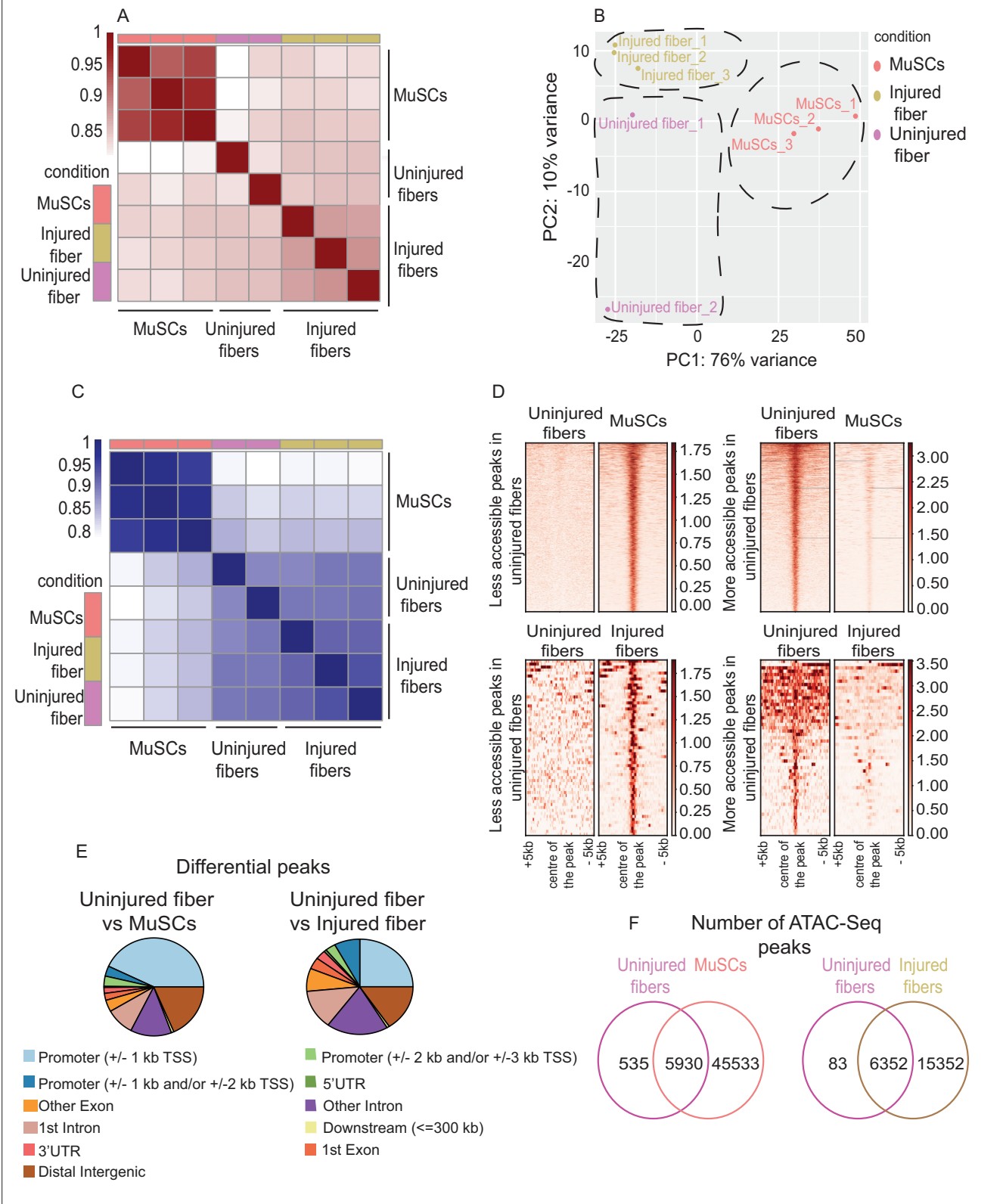

**Figure 3.** Uninjured and injured myofibers and MuSCs display distinct chromatin states. (**A**) Heatmap clustering of Pearson correlation coefficients showing the correlation between the replicates of the conditions in the regions defined by the union peakset (merged peaks of all replicates/samples). (**B**) Projection of samples along the first two principal components found by PCA showing the separate clustering of different samples and the clustering of each replicate of the same condition together. (**C**) Heatmap clustering of Pearson correlation coefficients indicating the correlation between the

*Figure 3 continued on next page*

*Figure 3 continued*

replicates in the regions defined by the consensus peakset derived from the uninjured myofibers, injured myofibers and MuSCs. (**D**) Pile-up analysis of differentially accessible peaks between uninjured myofibers and MuSCs and between injured myofibers and uninjured myofibers. Less accessible peaks: FDR < 0.05 and LFC < 0.5. More accessible peaks: FDR < 0.05 and LFC > 2. (**E**) Peak annotation pie charts for the differentially accessible peaks between uninjured myofibers vs MuSCs and uninjured myofibers vs injured myofibers. (**F**) Venn diagram of the number of ATAC-Seq peaks that are unique or overlapping between uninjured myofibers vs MuSCs and uninjured myofibers vs injured myofibers. *ATAC-Seq was performed in biological replicates (n = 3 MuSCs, n = 3 injured myofibers, n = 2 uninjured myofibers).

The online version of this article includes the following figure supplement(s) for figure 3:

**Figure supplement 1.** Correlation analysis between uninjured and injured myofibers only.

**Figure supplement 2.** IGV snapshots of genes expressed in fast and slow muscle fiber types.

**Figure supplement 3.** Unique peaks between different conditions indicate a distinct chromatin state for each cell type.

The unique chromatin state in each condition can be observed through the pile-up analysis that we have performed for the peaks identified as more accessible (LFC >2) and less accessible (LFC <0.5) between uninjured myofibers and MuSCs as well as between uninjured and injured myofibers (*Figure 3D*). In addition, the proportion of the differential peaks corresponding to various genomic regions, such as promoters and enhancers, differs depending on whether we compare MuSCs to myofibers or compare the myofibers during regeneration and homeostasis. For instance, the differential peaks between uninjured myofibers and MuSCs were mostly found close to the promoter region ( ≤ 1 kb), whereas in the uninjured and injured myofibers comparison, a greater proportion of differential peaks were found in the intron/distal intergenic regions (i.e. enhancer regions) (*Figure 3E*, *Table 5*). This implies that MuSCs and myofibers mostly differ in their promoter accessibility, whereas myofibers during homeostasis and regeneration differ mostly at the level of distal regulatory elements.

Furthermore, we performed occupancy analysis (using DiffBind) in order to determine the unique and common peaks between the conditions. Since the occupancy analysis relies on the peak score, the distribution pattern of the peak scores for all the conditions was assessed and were found to be similar despite the observed differences in the total number of peaks between the conditions (*Figure 3— figure supplement 3A*). Occupancy analysis between uninjured myofibers and MuSCs revealed that MuSCs have 45,533 unique peaks while myofibers contain only 535 unique peaks which are not present in MuSCs (*Figure 3F*). There are also many common accessible regions as seen from 5,930 peaks that are common to both uninjured myofibers and MuSCs. This analysis suggests that myonuclei share a large number of open chromatin regions with their parental stem cells. On the other hand, the occupancy analysis between uninjured and injured myofibers revealed that there are 6,352 overlapping peaks between the regenerating and resting myofibers. However, this analysis also revealed that there are 15,352 unique peaks in the injured myofibers and only 83 peaks that are unique to the resting myofiber (*Figure 3F*). Furthermore, when comparing the read count between conditions around the center of unique peaks, it can be observed that each condition displays a unique open chromatin signature (*Figure 3—figure supplement 3B,C*).

## Comparative analysis of the chromatin state between MuSCs and myofibers

To get a better understanding of the functional differences in chromatin accessibility between

**Table 5.** Percentage of differential peaks in each genomic feature.

| | Uninjured myofiber vs MuSCs (%) | Uninjured vs injured myofiber (%) | WT vs MDX myofiber (%) |
|---|---|---|---|
| Promoter (±1 kb TSS) | 43.07 | 25 | 29.92 |
| Promoter (±1 kb and/or ±2 kb TSS) | 3.36 | 7.81 | 3.68 |
| Promoter (±2 kb and/or ±3 kb TSS) | 3.39 | 3.12 | 4.49 |
| 5'UTR | 0.37 | 0.78 | 0.46 |
| 3'UTR | 1.95 | 3.12 | 2.99 |
| First Exon | 2.29 | 3.91 | 1.84 |
| Other Exon | 3.85 | 7.81 | 4.49 |
| First Intron | 9.16 | 13.28 | 14.84 |
| Other Intron | 13.34 | 18.75 | 24.86 |
| Downstream ( ≤ 300 kb) | 0.83 | 0.78 | 0.12 |
| Distal Intergenic | 18.39 | 15.62 | 12.31 |

MuSCs and myofibers, we first performed Gene Ontology (GO Biological Process) analysis on the genes associated with the nearest peaks from the uninjured myofiber and on the genes associated with the nearest unique peaks in the myofiber compared to MuSCs. As expected, this revealed myofiber-specific biological processes such as myofiber structure and organization (*Figure 4A* and *Figure 4—figure supplement 1A*). On the other hand, GO term analysis on all the genes nearest to each peak and on the genes associated with the nearest unique peaks to MuSCs revealed biological processes such as adherens junction organization, membrane permeability, and regulation of notch signaling which play key roles in MuSCs quiescence and function (*Bjornson et al., 2012*; *Mourikis and Tajbakhsh, 2014*) (*Figure 4B* and *Figure 4—figure supplement 1B*). The analysis above also revealed that genomic regions that remain in an open chromatin state when MuSCs fully differentiate into myofibers correspond to genes that are involved in processes such as mitochondrial transport, regulation of transcription, and regulation of metabolites and energy (*Figure 4—figure supplement 1C*). The changes in the chromatin state between MuSCs and myofibers can also be observed from the volcano plots showing differential peaks between conditions labeled by their nearest gene (*Figure 4D*). For instance, genomic regions associated with genes such as muscle-specific titin-capping protein (*Tcap*), a component of the skeletal muscle z-disc, as well as genes that are involved in regulatory and structural functions in skeletal muscle such as titin gene (*Ttn*) are associated with more accessible chromatin regions in the myofiber compared to MuSCs (*Markert et al., 2010*; *Hackman et al., 2002*) (*Figure 4D*).

Additionally, we performed GO term analysis between uninjured and injured myofibers, which revealed that globally, the accessible regions in the resting and regenerating myofibers corresponded to genes involved in similar processes. GO term analysis on the genes associated with the nearest peaks from uninjured myofibers and from the injured myofibers as well as the genes associated with the nearest peaks which are common between uninjured and injured myofibers revealed biological processes involved in striated muscle cell development, actomyosin structure, and sarcomere organization, which are important for myofiber structural formation and for the proper function of myofibers (*Figure 4A and C*, *Figure 4—figure supplement 1D*). On the other hand, genes associated with the nearest unique peaks from the injured myofibers mostly belong to processes involved in structural components of the myofiber while the genes associated with the nearest unique peaks from uninjured myofibers correspond to genes involved in ion transport and metabolism (*Figure 4—figure supplement 1E-F*).

Furthermore, we analyzed the enrichment of transcription factor binding motifs in the sequences under peaks common between the injured and uninjured myofibers overlapping the promoters (±5 kb of TSS) (*Figure 4—figure supplement 2A*) as well as in the peaks that are unique to injured and uninjured myofibers overlapping the promoters (±5 kb of TSS) (*Figure 4—figure supplement 2B*). The top motifs that were enriched in the sequences under peaks common to injured and uninjured myofibers include binding site for Mef2a (*Figure 4—figure supplement 2A*). On the other hand, the top motifs that were enriched in the sequences under peaks unique to injured myofibers included binding sites for JUN and Stat3 (*Figure 4—figure supplement 2B*). However, due to the low number of unique peaks in the uninjured myofibers (*Figure 3F*), there was no significant motif that enriched in that peak set.

## Identification of cell-type-specific pathways by global analysis of chromatin accessibility

To further understand the functional differences in chromatin accessibility between different cell types, we investigated the cell-type-specific pathways. To accomplish this, Gene Set Enrichment Analysis (GSEA) was performed on genes associated with differentially accessible peaks between the conditions. Importantly, the GSEA between uninjured and injured myofibers revealed that inflammatory response and Il2-Stat5 signaling, and injury related pathways are still operational even after 7 days of CTX-mediated injury to muscle (*Laurence et al., 2007*) (*Figure 5A*).

The GSEA between uninjured myofibers and MuSCs revealed that one of the significantly enriched pathways is myogenesis, where we can observe that genes associated with differentiation and myofiber function such as *Myf6*, *Ckm*,and Tropomyosin 2 (*Tpm2*) (*Lazure et al., 2020*; *Tai et al., 2011*; *Buckingham, 1994*; *Jin et al., 2016*) have higher accessibility in the myofiber compared to MuSCs (*Figure 5B-D*). On the other hand, genes associated with quiescence and MuSCs such as alpha seven

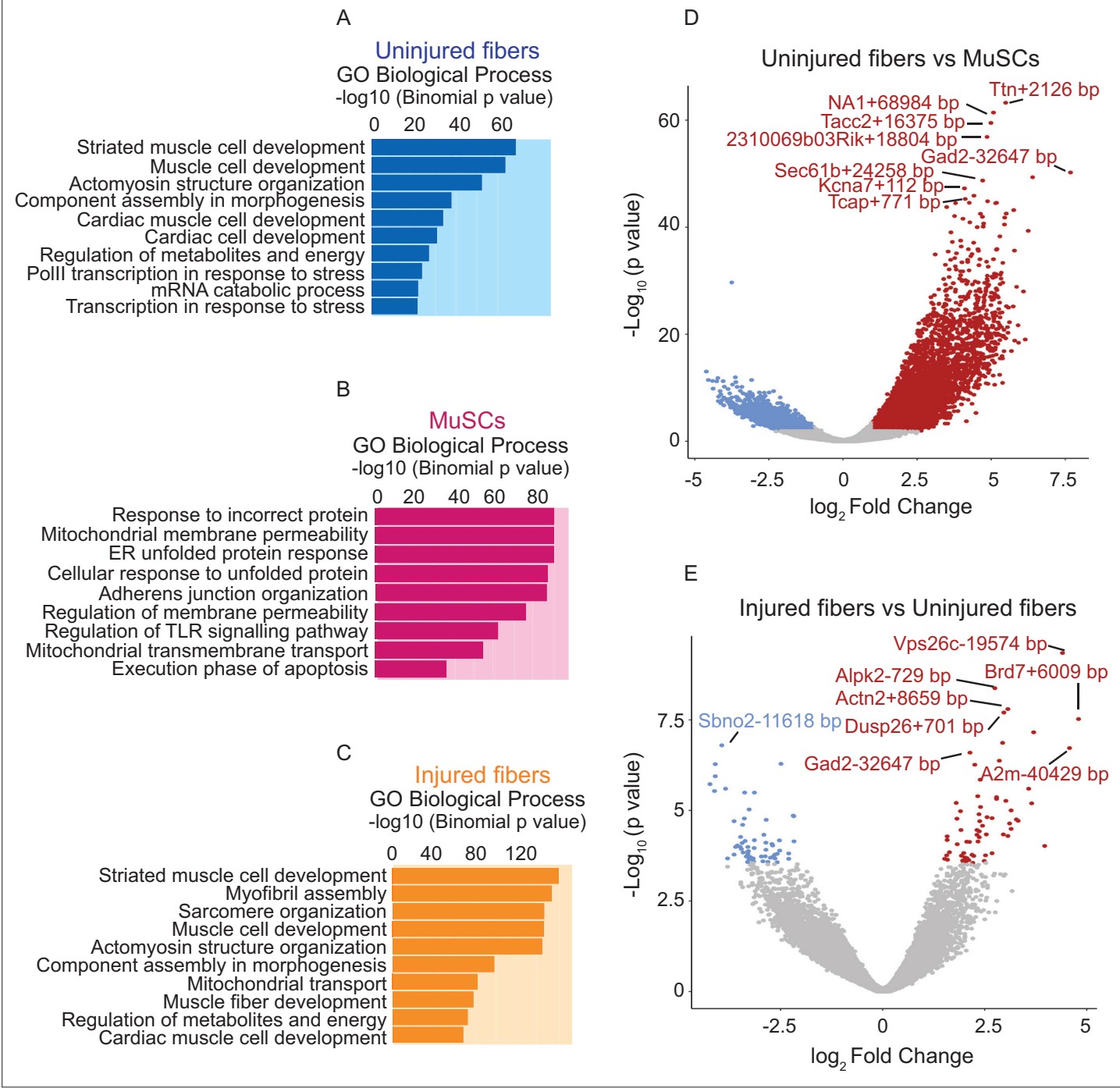

**Figure 4.** Comparative analysis of chromatin state between uninjured myofibers and MuSCs and between uninjured myofibers and injured myofibers. (A–C) Gene Ontology (GO Biological Process) analysis of genes associated with ATAC-Seq peaks based on association by proximity using Genomic Regions Enrichment of Annotations Tool (GREAT) (*McLean et al., 2010*) for all peaks present in the uninjured myofibers, MuSCs and injured myofibers, respectively. (D) Volcano plot of differentially accessible regions/peaks identified by FDR < 0.05 and LFC ≥ 1 between uninjured myofibers and MuSCs. Each dot represents a differentially accessible region/peak and the distance to the nearest gene is annotated. (E) Volcano plot of differentially accessible regions/peaks identified by FDR < 0.05 and LFC ≥ 1 between uninjured myofibers and injured myofibers. Each coloured dot represents a differentially accessible region/peak and the distance to the nearest gene is annotated. *ATAC-Seq was performed in biological replicates (n = 3 MuSCs, n = 3 injured myofibers, n = 2 uninjured myofibers).

The online version of this article includes the following figure supplement(s) for figure 4:

**Figure supplement 1.** Gene Ontology analysis of unique and common peaks between conditions.

**Figure supplement 2.** Top enriched motifs in the ATAC-Seq peaks of uninjured and injured myofibers.

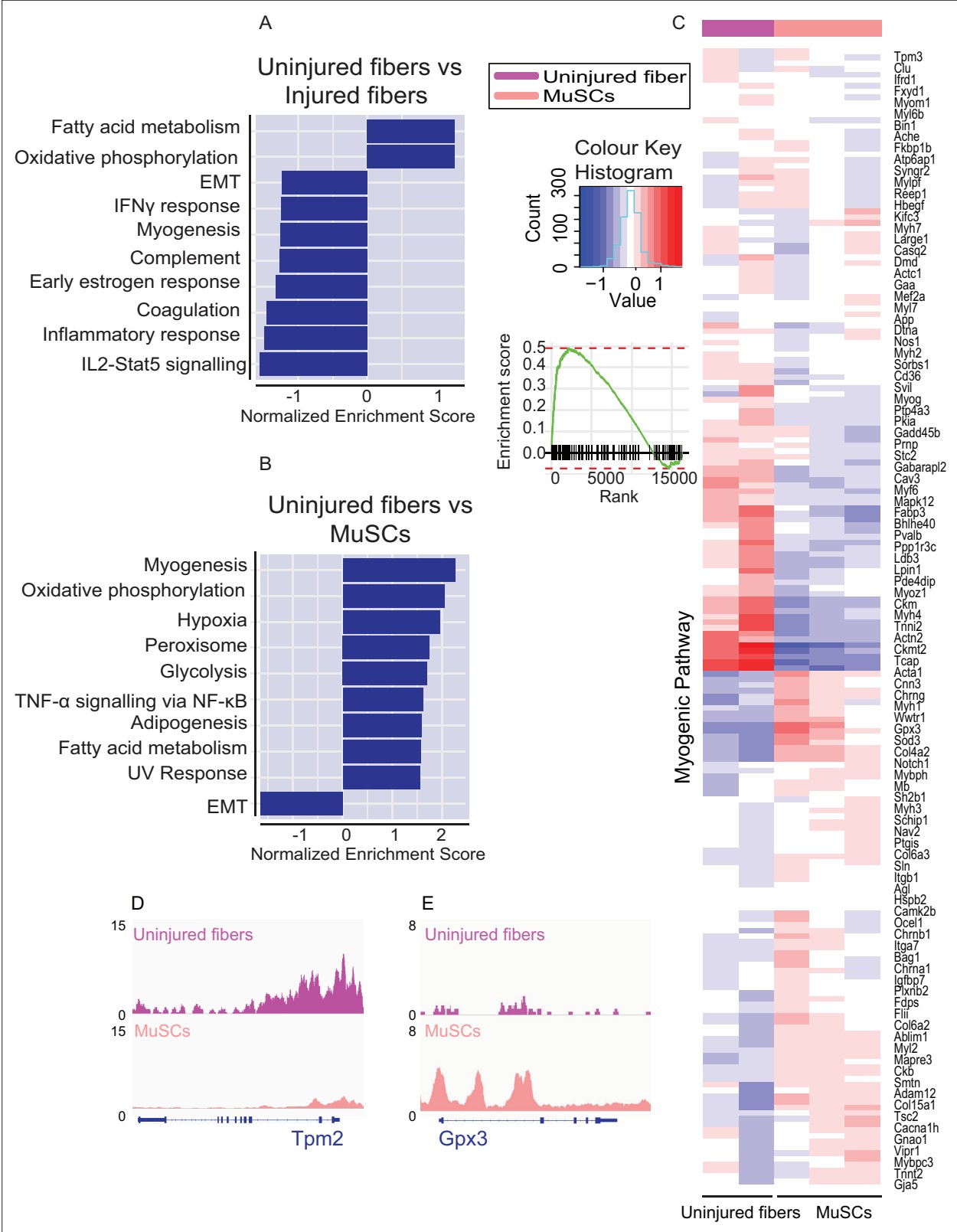

**Figure 5.** Identification of cell type specific pathways by global analysis of chromatin accessibility. (**A**) Gene Set Enrichment Analysis performed on genes nearest to the differentially accessible regions/peaks for uninjured myofibers compared to injured myofibers. Top 10 enriched pathways are shown although do not reach significance. (**B**) Gene Set Enrichment Analysis performed on genes nearest to the differentially accessible regions/peaks for uninjured fibers compared to MuSCs. Top 10 significantly enriched pathways are shown (FDR < 0.01). (**C**) Heatmap for genes involved in myogenesis

*Figure 5 continued on next page*

*Figure 5 continued*

based on read counts of MuSCs and uninjured fibers ±1 kb of the TSS of each gene in the myogenic pathway. (**D**) IGV snapshot of Tropomyosin 2 (*Tpm2*). (**E**) IGV snapshot of Glutathione Peroxidase 3 (*Gpx3*). *ATAC-Seq was performed in biological replicates (n = 3 MuSCs, n = 3 injured myofibers, n = 2 uninjured myofibers).

The online version of this article includes the following figure supplement(s) for figure 5:

**Figure supplement 1.** Analysis of Notch and TGFβ signalling pathways reveal differential accessibility between MuSCS and uninjured myofibers, and injured and uninjured myofibers.

integrin (*Itga7*) and *Gpx3* are more accessible in MuSCs compared to the myofiber, as expected (*El Haddad et al., 2012*; *Pasut et al., 2012*) (*Figure 5C and E*).

Moreover, differential chromatin accessibility between the MuSCs and their myofiber derivatives show differences in pathways that are known to be important for muscle, such as Notch and TGFβ signalling (*Bjornson et al., 2012*; *Mourikis and Tajbakhsh, 2014*; *Carlson et al., 2009*; *Girardi et al., 2021*) (*Figure 5—figure supplement 1*). For example, increased accessibility of *Notch1* is seen in MuSCs while increased accessibility of Jagged-2 (*Jag2*) is observed in the myofibers regardless of whether they are regenerating or homeostatic as seen by the height of the peaks at their promoters (*Figure 5—figure supplement 1A-C*). On the other hand, for TGFβ signalling, Noggin (*Nog*) shows more accessibility in the myofibers while bone morphogenetic protein-4 (*Bmp4*) has increased accessibility in MuSCs (*Figure 5—figure supplement 1D-F*). Taken together, this data shows that smfATAC-Seq is an effective method to analyze chromatin accessibility and to identify active cis-regulatory elements in a single muscle fiber as well as to compare muscle fibers under different physiological conditions.

## Comparative analysis of the cromatin state between WT and MDX myofibers

To demonstrate the applicability of our method to a disease condition, we performed smfATAC-Seq on myofibers isolated from mdx mice, a model for Duchenne's muscular dystrophy (DMD), and their WT C57BL/10ScSn counterparts. In order to solely assess the effect of the loss of dystrophin without the effect of regeneration, myofibers that were not actively regenerating were selected for processing from both conditions. As was performed in the previous cohort, we began by confirming the similarity between biological replicates by visualization of ATAC-Seq peaks for *Ckm* and housekeeping gene *Rps2*, as well by the Pearson correlation analysis between the biological replicates (*Figure 6—figure supplement 1*). After consistency of the samples within the conditions was established, the biological replicates from the same condition were pooled for further analysis. First, enrichment of the ATAC-Seq reads around the TSS from both mdx and WT myofibers was confirmed (*Figure 6A*). Peak annotation analysis revealed that most of the peaks were in the promoter and enhancer regions for both data sets (*Figure 6B*, *Table 4*). To further show that smf-ATAC-Seq can successfully assess the chromatin accessibility in a single myofiber of an mdx and WT EDL muscle, we looked at the presence of ATAC-seq peaks in the promoter regions of *Ckm*, *Acta1*, *Myh4,* and the housekeeping genes *Gapdh* and *Rps2*, whereas *Pou5f1* was used as a negative control (*Figure 6—figure supplement 2A-F*). We also confirmed that the myofibers from mdx and WT conditions were fast type (*Figure 6—figure supplement 2G-H*) and that they exclusively represent the myonuclei, without the presence of confounding cell types in the muscle (*Figure 6—figure supplement 2I-N*).

To investigate the differences in the chromatin state between mdx and WT, we first performed heatmap clustering of Pearson correlation coefficients and PCA (*Figure 6C and D*). These analyses showed that, mdx and WT myofibers cluster separately and the biological replicates for each condition generally cluster together (*Figure 6C and D*). We then performed differential analysis of ATAC-Seq peaks followed by pile-up analysis for the less accessible (LFC <0.5) and more accessible (LFC >2) peaks between WT and mdx myofibers (*Figure 6E*). The results clearly showed that mdx and WT myofibers exhibit extensive differences in their chromatin states (*Figure 6E*). Peak annotation analysis on the differential peaks between mdx and WT myofibers revealed that more than half of the differential peaks were found in enhancer regions, indicating that they differ mostly at the level of distal regulatory elements (*Figure 6F*, *Table 5*). In addition, occupancy analysis revealed that there

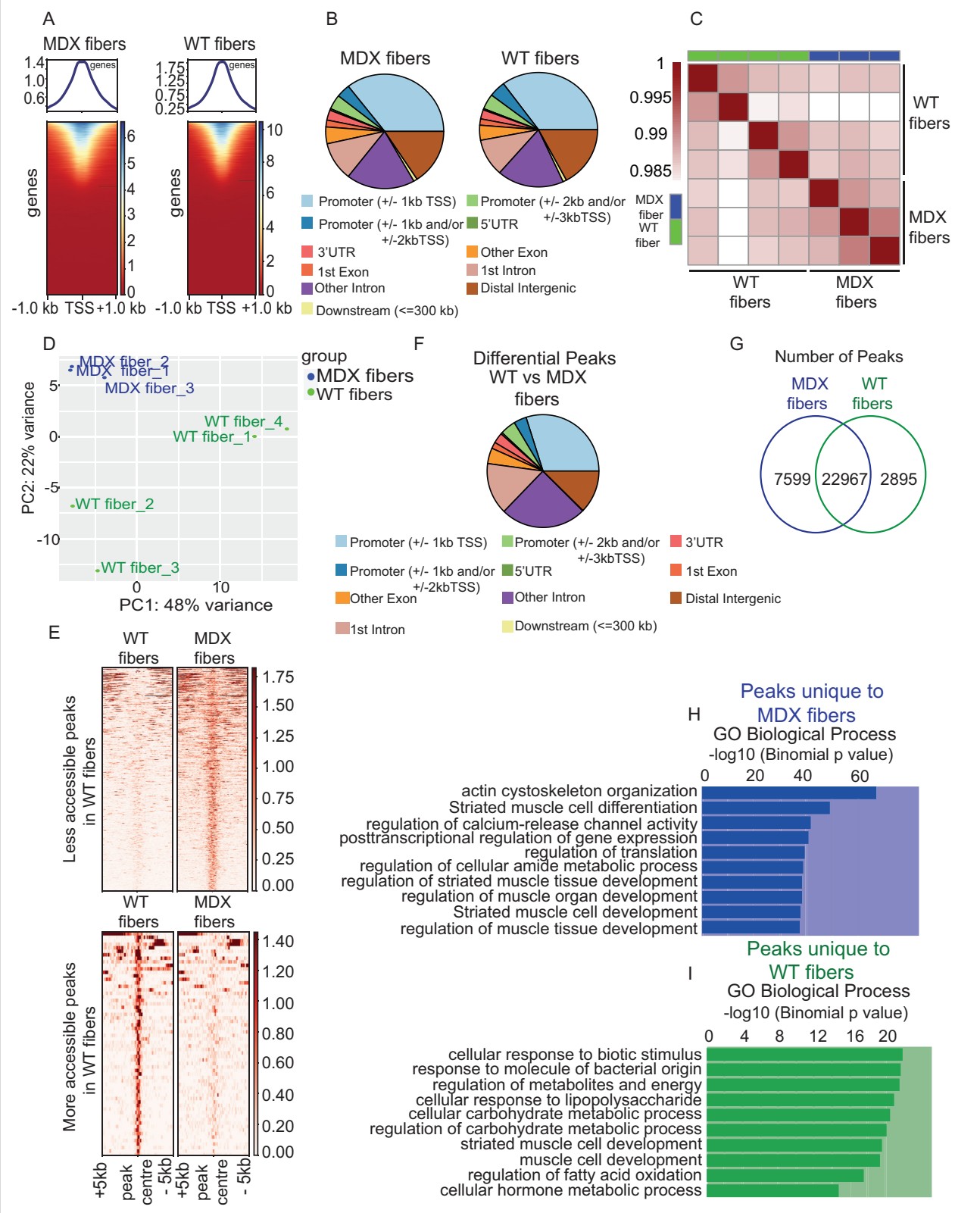

**Figure 6.** Comparative analysis of chromatin state between MDX and WT myofibers. (**A**) Heatmaps showing enrichment at transcription start site (TSS) for the ATAC-Seq libraries of MDX and WT myofibers respectively. (**B**) Peak annotation pie charts for ATAC-Seq peaks of MDX and WT myofibers respectively. (**C**) Heatmap clustering of Pearson correlation coefficients showing the correlation between the replicates of the conditions in the regions defined by the union peakset (merged peaks of all replicates/samples). (**D**) Projection of samples along first two principal components found by

*Figure 6 continued on next page*

*Figure 6 continued*

PCA showing the separate clustering of different samples and the clustering of each replicate of the same condition together. (**E**) Pile-up analysis of differentially accessible peaks between WT and MDX myofibers. Less accessible regions: FDR < 0.05 and LFC < 0.5. More accessible peaks: FDR < 0.05 and LFC > 2. (**F**) Peak annotation pie charts for the differentially accessible peaks between WT and MDX myofibers. (**G**) Venn diagram of the number of ATAC-Seq peaks that are unique or overlapping between WT and MDX myofibers. (**H**) Gene Ontology (GO Biological Process) analysis of genes associated with unique peaks present in the MDX myofiber compared to WT myofibers, based on the proximity of the peaks to the genes. (**I**) Gene Ontology (GO Biological Process) analysis of genes associated with unique peaks present in the WT myofiber compared to MDX. *ATAC-Seq on the myofibers were performed in biological replicates (n = 3 MDX myofibers, n = 4 WT myofibers).

The online version of this article includes the following figure supplement(s) for figure 6:

**Figure supplement 1.** Correlation analysis between biological replicates of mdx and WT myofiber ATAC-Seq samples.

**Figure supplement 2.** IGV snapshots of myogenic and non-myogenic genes for the quality control of mdx and WT smfATAC-Seq.

**Figure supplement 3.** Gene Ontology analysis of total mdx and WT peaks.

**Figure supplement 4.** Correlation analysis between injured, mdx and WT myofibers.

**Figure supplement 5.** Top enriched motifs in the ATAC-Seq peaks of mdx and WT myofibers.

are 22,967 overlapping peaks between mdx and WT myofibers, and that mdx myofibers possess 7599 unique peaks, while WT myofibers contain 2895 unique peaks (*Figure 6G*).

Furthermore, to understand the functional differences in chromatin accessibility, we performed Gene Ontology (GO Biological Process) analysis on the genes associated with the nearest peak for all peaks in the mdx and WT myofibers, as well as on the genes associated with the nearest common peaks between them (*Figure 6—figure supplement 3*). These analyses revealed processes involved in mitochondrial transport, myofibril assembly, sarcomere organization, and striated muscle cell development, which are important for myofiber structure, organization, and function (*Figure 6—figure supplement 3*). However, GO term analysis on the unique peaks of each condition revealed that different processes are affected between mdx and WT. GO term analysis on the genes associated with the nearest unique peaks from mdx myofibers revealed processes that are important for myofiber structure and organization such as actin cytoskeleton organization and striated muscle cell differentiation (*Figure 6H*). On the other hand, GO term analysis on the genes associated with the nearest unique peaks from WT myofibers revealed processes mostly involved in metabolism (*Figure 6I*). Since the observed differential biological processes between WT and mdx myofibers were similar to those seen between injured vs uninjured myofibers, we then compared the overall differences in chromatin accessibility between mdx, WT, and injured myofibers. We performed heatmap clustering of Pearson correlation and PCA analysis between WT, mdx and injured myofibers (*Figure 6—figure supplement 4*). These analyses have revealed that injured myofibers were more similar to the mdx than they are to the WT C57BL/10ScSn myofibers. However, as expected due to the different genetic backgrounds of the mice between injured and the WT and mdx mice, WT and mdx myofibers were more similar to each other than they are to the injured C57BL/6 myofibers (*Figure 6—figure supplement 4*).

Finally, we determined the top motifs that are enriched in the sequences under the peaks that are common between the mdx and WT myofibers overlapping the promoters (±5 kb of TSS) (*Figure 6—figure supplement 5A*) as well as the sequences under peaks that are unique to mdx and WT overlapping the promoters (±5 kb of TSS) (*Figure 6—figure supplement 5B-C*). The top significantly enriched motifs in the peaks common between mdx and WT included Mef2a and JUN (*Figure 6—figure supplement 5A*) while the top motifs enriched in the peaks unique to mdx included transcription factors such as Foxo1 (*Figure 6—figure supplement 5B*).

Overall, this data shows that smfATAC-Seq can be reliably used to study myofibers in disease conditions, revealing that there are substantial differences in the chromatin accessibility of myofibers in the mdx mice compared to their WT counterparts.

## Discussion

Analysis of the myofiber-specific chromatin state and gene expression profile is very limited due to the heterogenous nature of muscle with the presence of numerous non-myogenic cells in the tissue. Whole muscle or muscle biopsies represent a pooled result of numerous cell types which are present in the muscle tissue. To overcome this limitation, single nucleus RNA-Seq (snRNA-Seq) and single

nucleus ATAC-Seq (snATAC-Seq) have been developed and performed on myonuclei to allow for the computational removal of other cell types (*Petrany et al., 2020*; *Dos Santos et al., 2020*; *Kim et al., 2020*). However, these methods still sequence all myonuclei present in the muscle and cannot distinguish between different myofibers within a muscle. Although snATAC-Seq provides high dimensionality, it is limited in sequencing depth due to the generation of sparse reads. Although computational pseudo bulking of snATAC-Seq can increase read numbers for comparative analysis between samples and conditions, the pooled reads represent the average of all myonuclei within the sample.

In this study, we have introduced a highly effective protocol based on the adaption of OMNI ATAC-Seq (*Corces et al., 2017*) to quantify chromatin accessibility of a single EDL myofiber with high resolution and sequencing depth. This method allows for comparative analysis of chromatin accessibility within and between muscle types with a potential for wide-spread use in future studies to investigate myofiber-specific epigenetic alterations in skeletal muscle.

The smfATAC-Seq protocol that we introduce in this study investigates the open chromatin state of myofibers at a single myofiber resolution, and with a high sequencing depth that allows for peak calling and differential peak analysis. Using this method, we have demonstrated that accessible chromatin regions of myonuclei contained within a single EDL myofiber can be tagmented and that high-quality sequencing ready libraries can be generated from these fragments. Sequencing of these libraries allow for sufficient depth and peak calling that can be used for genome-wide analysis of chromatin accessibility between myofibers. In this study, we have also demonstrated that smfATAC-Seq strictly investigates a single myofiber without the confounding presence of muscle resident non-myogenic cell types. Additionally, application of trypsin to the isolated myofibers effectively removes MuSCs that are associated with myofibers (*Blackburn et al., 2019*) and was confirmed by the absence of peaks at the promoters of known genes associated with muscle stem and niche cells. Although all the smfATAC-Seq samples sequenced in this study were fast type myofibers, this protocol can be used to distinguish between different fiber types which could be applied to study myofiber heterogeneity under various physiological conditions. smfATAC-Seq peaks associated with genes involved in muscle structure and function such as *Acta1*, *Ckm* and the myosin heavy chain cluster, indicates that our smfATAC-Seq is a robust technique to investigate genome-wide chromatin accessibility of a single myofiber.

A key implication of this technique is its applicability to the study of changes in chromatin accessibility between myofibers in different contexts. As a demonstration of this, we have performed smfATAC-Seq on uninjured myofibers as well as injured myofibers isolated 7 days post-injury to investigate the changes in chromatin accessibility that occurs during regeneration. Through Pearson correlations and PCA analysis, we showed resting and injured myofibers cluster separately, indicating the power of smfATAC-Seq to determine chromatin signature from even the minute starting material of a single myofiber. In addition, through occupancy analysis, we showed that there is a large difference in the number of unique peaks present in the injured myofibers compared to uninjured myofibers. This indicates that there are major modifications to chromatin accessibility in the context of regeneration. However, GO term analysis of genes associated with the accessible chromatin regions in both injured and uninjured fibers are very similar in the biological processes and pathways that are enriched such as striated muscle cell development, actomyosin structure and sarcomere organization, which are key factors for the proper structure and function of muscle. Despite these similarities, there are certain trends in which uninjured myofibers have increased accessibility in genes involved in energy metabolism, while injured myofibers have greater accessibility in genes involved in myogenesis and inflammatory response which is what would be expected in the case of an injury and regeneration (*Tidball, 2011*). Despite the increase in chromatin accessibility during injury, the accessible chromatin regions in both injured and uninjured fibers are associated with genes involved in similar biological processes. This similarity at the gene network despite differences in chromatin profile may suggest activation of multiple enhancers on core muscle structural genes in the case of injury. Another possible reason could be the length of the recovery time where at 7 days post injury, a number of genes activated early in the regeneration process may have returned to levels seen in the steady state. Previously, a study investigating the changes in the transcriptional profile of MuSCs and various muscle resident cells throughout different time points of muscle injury using single cell RNA-Seq, revealed that after seven days of regeneration most cell types returned to a state that was similar to homeostasis (*De Micheli et al., 2020a*). Therefore, it is possible that harvesting the injured EDL myofibers 7 days post

injury allowed these myofibers to return to a state reminiscent of homeostatic myofibers. Further, our analyses of the myofibers in this study indicates that this technique can effectively compare samples between conditions and could see future use in the study of chromatin accessibility of myofibers under different biologically relevant conditions.

We have also used smfATAC-Seq to compare changes in chromatin accessibility between MuSCs and myofibers. Our data shows that the regions of open chromatin in the myofibers correspond to genes involved in structural components of the muscle, such as the z-disc, which are important for the proper functioning of the muscle. On the other hand, open regions of chromatin in the MuSCs mostly correspond to genes involved in membrane permeability, adherens junction organization and signalling pathways implicated in the regulation of MuSC function (*Relaix et al., 2021*). The analysis also revealed that chromatin regions that are accessible in both MuSCs and myofibers correspond to genes that are crucial for the general function of cells such as those involved in mitochondrial transport, regulation of transcription, and regulation of metabolites and energy.

Lastly, we performed smfATAC-Seq on non-regenerating myofibers isolated from the mouse model of DMD (*McGreevy et al., 2015*) and their WT counterparts. DMD is a type of muscular dystrophy caused by a loss-of-function mutation in the Dystrophin (*DMD*) gene that encodes for a protein that has a crucial role in muscle structure (*Gao and McNally, 2015*). Lack of functional dystrophin in DMD leads to unstable and fragile myofibers that continuously need to be regenerated, in turn leading to progressive muscle degeneration (*Gao and McNally, 2015*). We have not only shown that smfATAC-Seq can reliably investigate the chromatin accessibility from a single myofiber of mdx and WT EDL muscle, but through Pearson correlations, PCA and differential peak analysis we have also shown that DMD is associated with substantial alterations in chromatin accessibility in myonuclei. Through occupancy and GO term analyses, we have shown that a great proportion of peaks that are common between the mdx and WT myofibers are mostly associated with processes involved in muscle structure and organization. However, we have shown that mdx myofibers have more unique peaks compared to the WT, suggesting that chromatin accessibility of myonuclei is increased in the mdx disease model. Our analyses have shown that the unique peaks in mdx are associated with biological processes involved in myofiber structure and organization. On the other hand, unique peaks in the WT myofibers are associated with processes mostly involved in energy and metabolism. It is possible that the progressive muscle degeneration due to loss of muscle fiber integrity and stability causes mdx myofibers to compensate by increasing the activity of processes involved in muscle structure and organization while WT myofibers retain their activity in metabolism. It should be noted that the differences in chromatin accessibility that we have observed between resting and regenerating myofibers and between the mdx and WT myofibers are similar, which could be explained by the degeneration and continuous round of regeneration in the mdx myofibers.

Studies in the future could utilize smfATAC-Seq to further investigate the changes in chromatin accessibility in the mdx myofibers and investigate the changes associated with DMD at the level of chromatin to potentially investigate therapeutic avenues for this disease, as well as other muscle wasting diseases.

Overall, smfATAC-Seq is a robust molecular tool that can be used to analyze genome-wide chromatin accessibility of a single myofiber. The sequencing depth from this approach, allows for in-depth analysis, peak calling, quantitative analysis of chromatin accessibility and to identify active enhancers and promoters in a single muscle fiber. smfATAC-Seq can be used to study the epigenetic alterations that occur in muscle fibers during development, diseases, and in response to exercise.

# Materials and methods

**Key resources table**

| Reagent type (species) or resource | Designation | Source or reference | Identifiers | Additional information |
|---|---|---|---|---|
| Genetic reagent (*M. musculus*) | C57BL/6 J | The Jackson Laboratory | Stock #: 000664 | |
| Genetic reagent (*M. musculus*) | C57BL/10ScSnJ | The Jackson Laboratory | Stock #: 000476 | |

*Continued on next page*

*Continued*

| Reagent type (species) or resource | Designation | Source or reference | Identifiers | Additional information |
|---|---|---|---|---|
| Genetic reagent (*M. musculus*) | C57BL/10ScSn-*Dmd*mdx/J | The Jackson Laboratory | Stock #: 001801 | |
| Genetic reagent (*M. musculus*) | Tg(Pax7-EGFE)#Tagb (Pax7-nGFP) | Sambasivan, R. et al. Distinct Regulatory Cascades Govern Extraocular and Pharyngeal Arch Muscle Progenitor Cell Fates. Developmental Cell, (2009). (*Sambasivan et al., 2009*) | PMID:19531352 | Dr. Shahragim Tajbakhsh (Institut Pasteur) |
| Commercial kit or assay | Tn5 transposase | Illumina | Cat #: 20034197 | |
| Commercial kit or assay | Nextera XT adaptors | Illumina | Cat #: FC-131–1001 | |
| Commercial kit or assay | QIAquick PCR purification kit | Qiagen | Cat #: 28,104 | |
| Chemical compound, drug | Triton X –100 | Sigma-Aldrich | Cat #: T9284 | |
| Chemical compound, drug | Tween-20 | Sigma-Aldrich | Cat #: P1379-1L | |
| Chemical compound, drug | Digitonin | Promega | Cat #: G9441 | |
| Chemical compound, drug | Collagenase D | Roche | Cat #: 11088882001 | 2.4 U/mL |
| Chemical compound, drug | Collagenase | Sigma-Aldrich | Cat #: C0130 | 1000 U/mL |
| Chemical compound, drug | Dispase II | Roche | Cat #: 39307800 | 12 U/mL |
| Chemical compound, drug | Cardiotoxin | Sigma Aldrich | Cat #: 11061-96-4 | |
| Sequence-based reagent | MyoD_L | This paper | PCR primers | TGCTCCTTTG AGACAGCAGA |
| Sequence-based reagent | MyoD_R | This paper | PCR primers | AGTAGGGAA GTGTGCGTGCT |
| Other | Q5 High Fidelity DNA polymerase | New England Biolabs | Cat #: M0491S | For amplification of DNA post Tn5 tagmentation (see Library Preparation) |
| Chemical compound | DAPI stain | Invitrogen | Cat #: D3671 | (5 mg/mL) |
| Other | Ampure XP beads | Beckman | Cat #: A63880 | For library size selection at a concentration of 0.85 x (see Library Preparation) |
| Chemical compound | Hoechst | Molecular Probes | Cat #: H1399 | (5 mg/mL) |

## ATAC-Seq on a single myofiber

### Isolation of Extensor Digitorum Longus (EDL) from cardiotoxin-induced injured muscle

The Extensor Digitorum Longus (EDL) muscle was injured by intramuscular injection of 50 µL of 5 µM cardiotoxin (CTX) (Sigma, 11061-96-4). Mice were treated with carprofen 20 minutes prior to CTX injection and were injected with CTX under anesthesia by isoflurane. Mice were sacrificed 7 days post injury and the EDL was collected from the hind limb of each mouse with the contra lateral EDL being used for the isolation of uninjured myofibers.

### Dissection of EDL muscle

The EDL muscle was dissected as previously described (*Blackburn et al., 2019*). Briefly, the skin of the hindlimb was removed and the tibialis anterior (TA) muscle was excised with a pair of dissection

scissors. The tendons of the EDL were exposed and the EDL was cut from tendon to tendon with scissors.

## Isolation of a single EDL myofiber

Individual myofibers were isolated from the EDL muscle as previously described (*Blackburn et al., 2019*). Briefly, the intact EDL muscle was placed in a 1.5 mL eppendorf tube with 800 µL of myofiber digestion buffer containing 1000 U/mL of collagenase from *Clostridium histolyticum* (Sigma, C0130) in un-supplemented DMEM (Gibco, 11995–065) for 1 hr. Trypsin was added to the myofiber digestion buffer at a final concentration of 0.25% to remove the myofiber associated muscle stem cells. The EDL myofibers were then transferred into 2 mL of 1 X PBS (Wisent, 311–425 CL) in a six well-plate that had previously been coated with DMEM supplemented with 10% horse serum (HS) (Wisent, 065250). The EDL was then gently pipetted up and down with a large-bore glass pipette to disassociate the myofibers.

## Selection of injured and uninjured myofibers

Live myofibers in the six-well plate were stained with 2 µL of 5 mg/mL of Hoechst (Molecular Probes, H1399) in 2 mL of 1 X PBS for 5 minutes in a 37 °C with 5% $CO_2$ incubator. The myofibers were then visualized under a microscope in DAPI channel and selected based on the myonuclei location, where myofibers with a pattern of centrally located nuclei were determined to be regenerating and picked for the injury condition. Individual myofibers were then transferred to 0.2 mL microtubes using a small-bore glass pipette coated with HS.

## Lysis and permeabilization of the myofiber

Residual media was removed with a pipette under a microscope. Individual myofibers in 0.2 mL micro-tubes were put in 10 µL of $ddH_2O$ for 5 min on ice. The $ddH_2O$ was removed with a pipette under a microscope, ensuring that the myofiber remained in the tube. The myofiber was then permeabilized with 20 µL of 0.5% Triton X-100 (Sigma, T9284) in PBS for 15 min at room temperature (RT). The permeabilization buffer was removed with a pipette under a microscope and the myofiber was washed twice with 200 µL of 1 X PBS.

## Tagmentation of the myofiber by Tn5 transposase

Transposition and ATAC-seq library preparation for a single myofiber was adapted from previously described OMNI ATAC-Seq protocol (*Corces et al., 2017*). The permeabilized myonuclei were tagmented with tagmentation mixture optimized for use on a myofiber (20 µL Tagment DNA Buffer (TD Buffer) (Illumina, 20034197), 13.3 µL PBS, 0.2% Tween-20 (Sigma, P1379-1L), 0.02% Digitonin (Promega, G9441), 1.39 µL Tn5 (Illumina, 20034197) and 4.61 µL water). Each single myofiber was incubated with 6 µL of the tagmentation mixture at 37 °C for 56 minutes with periodic shaking of the tubes every 5–7 minutes. Following the transposition with Tn5, DNA was purified using a QIAquick PCR Purification Kit (Qiagen, 28104) according to the manufacturer's guidelines.

## Library preparation

The purified DNA was PCR amplified for 15 cycles using Q5 High Fidelity DNA polymerase (New England Biolabs, M0491S) with the incorporation of Illumina Nextera XT adaptors (Illumina, FC-131–1001). The libraries were then size selected with AmpureXP Beads (Beckman, Cat# A63880) at a 1: 0.85 ratio (v/v). The size selected libraries were verified for quality control by bioanalyzer as well as verification of the library size via visualization on an agarose gel stained with GelGreen dye (Biotium, 41005). Libraries were then sequenced on NovaSeq6000 Sprime Paired End (PE) 150 bp.

## ATAC-Seq on MuSCs

### Isolation of MuSCs by fluorescence-activated cell sorting (FACS) for ATAC-Seq

MuSCs were isolated by Fluorescence Activated Cell Sorting (FACS) as previously described (*Tichy et al., 2018*). Briefly, hindlimb muscles from Pax7/GFP+ mice were dissected and chopped. The minced muscles were then transferred into a 15 mL Falcon tube and digested in un-supplemented F10 media (Gibco, 11550043) with 2.4 U/mL Collagenase D (Roche, 11088882001), 12 U/mL Dispase II (Roche, 39307800), and 0.5 mM $CaCl_2$. Digestion was performed on a shaker in an incubator at 37 °C with

5% $CO_2$ for 30 min. Following the first digestion, digested muscles were centrifuged at 600 g for 20 s and the supernatant was transferred to a 50 mL Falcon tube with 9 mL FBS (Wisent, 080450) and was kept on ice. The remaining pellet was triturated and was digested for another 15 min with additional digestion buffer added. After the final digestion, the digested muscle mixture was transferred to the 50 ml Falcon tube containing the previously digested mixture. The digested muscle mixture was then filtered through a 40-μm cell strainer (Falcon, C352340) and was centrifuged at 600 g for 18 min at 4 °C. The pelleted cells were then resuspended in 800 μL FACS buffer that is composed of 2% FBS/PBS (v:v), 0.5 mM EDTA (Invitrogen, AM9261) and with 0.5 μL DAPI (5 mg/mL) (Invitrogen, D3671). Resuspended cells were then filtered through 40-μm cell strainer and were transferred into polypropylene round-bottom FACS compatible tubes (Falcon, 352063). MuSCs were sorted with a FACSAria Fusion cytometer (BD Biosciences) based on negative selection for DAPI and positive selection for GFP.

## Lysis and transposition of MuSCs

ATAC-Seq on MuSCs was performed based on the previously established OMNI-ATAC-Seq protocol (*Corces et al., 2017*). Briefly, five thousand MuSCs were sorted by FACS into 30 μL of the ATAC lysis buffer containing 10 mM Tris-HCl (pH 7.5), 10 mM NaCl (Bioshop, 7647-14-5), 3 mM MgCl2 (Sigma, 7786-30-3), 0.1% Tween-20 (Sigma, P1379-1L), 0.1% NP-40 (Sigma, 74385), and 0.01% Digitonin (Promega, G9441) in a 0.2 mL microtube. Cells were incubated in the lysis buffer for 5 min on ice and then 3 min at room temperature (RT). Cells were then washed with 100 μL of wash buffer composed of 10 mM Tris-HCl (pH 7.5), 10 Mm NaCl, 3 mM MgCl2 and 0.1% Tween-20, and were centrifuged at 800 g for 10 min. The pellet was resuspended in 10 μL of transposition mixture (5 μL TD buffer, 3.2 μL PBS, 0.89 μL Tn5 (Illumina, 20034197), 0.1% Tween-20, 0.01% Digitonin and 0.75 μL nuclease free water). Transposition was performed for 20 min at 37 °C while shaking the tubes every 5–7 min. The DNA was then purified using a QIAquick PCR Purification Kit according to the manufacturer's guidelines.

## Library preparation for MuSCs ATAC-Seq

The eluted tagmented DNA was PCR amplified for 12 cycles with the incorporation of Illumina Nextera XT adapters using Q5 High Fidelity DNA polymerase. The libraries were then size selected with AmpureXP Beads at a 1: 0.85 ratio (v/v). The libraries were then verified by bioanalyzer and agarose gel visualization. Finally, the samples were sequenced on NovaSeq6000 Sprime Paired End (PE) 150 bp.

## ATAC-Seq data processing

The sequencing data was processed using the GenPipes pipeline v.3.1.5 (*Bourgey et al., 2019*). The raw reads were trimmed using Trimmomatic v.0.36 (*Bolger et al., 2014*) and aligned to the mm10 genome assembly using the Burrows-Wheeler Aligner v.0.7.12 (*Li and Durbin, 2009*). Reads were filtered to keep only high quality alignments (MAPQ score >20) and duplicates were removed using SAMtools v.1.3.1 (*Li et al., 2009*). Peak calling was performed with MACS2 v.2.1.1 (*Zhang et al., 2008*) using piling up of paired-end fragment mode (*--format BAMPE*). The peak files (bed) were filtered by removing the ENCODE black listed regions (https://www.encodeproject.org/files/ENCFF547MET) using BEDTools v2.29.1 (*Quinlan, 2014*). Mitochondrial reads were also removed before the analysis.

## Correlation analysis between the biological replicates and clustering

In order to perform a quantitative comparison of the read counts within accessible regions, the overlapping peaks of all replicates were merged using BEDTools v2.29.1 (*Quinlan, 2014*). This set of merged peaks and the BAM alignment files were used as input for the *featureCounts* function of Rsubread v.2.2.6 (*Liao et al., 2014*) to generate a raw-count matrix. The raw counts were normalized by rlog transformation using DESeq2 (*Love et al., 2014*) with respect to library size. Pearson correlation coefficients were calculated based on the normalized counts for each pairwise comparison. Principal component analysis (PCA) and hierarchical clustering were also performed to evaluate the similarity between the replicates.

## Peak annotation analysis

For each condition, the BAM alignment files of the replicates were merged and peak calling was performed with MACS2 v.2.1.1 (*Zhang et al., 2008*). Peak sets for each condition were annotated using the ChIPseaker v.1.24.0 (*Yu et al., 2015*) *annotatePeak* function, and the UCSC Genome Browser knownGene (mm10) table.

## Obtaining coverage tracks

The BAM alignment files were converted to bigWig format and normalized by scaling factor (--scaleFactor) with the deepTools v.2.5.0.1 (*Ramírez et al., 2014*) *bamCoverage* function.

## Enrichment of genomic signal around TSS

The bigWig files and the TSS coordinates obtained from the UCSC Genome Browser knownGene (mm10) table were used as input for the *computeMatrix* function of deepTools v.2.5.0.1 (*Ramírez et al., 2014*). This matrix was used for *plotHeatmap* function to generate the heatmap.

## Identification of overlapping/unique accessible regions

For each comparison between the conditions, overlapping and unique accessible regions were identified with DiffBind v.2.16.2 (*Stark, 2011*) based on the measure of confidence in the peak call by MACS2 v.2.1.1 (*Zhang et al., 2008*).

## Analysis of differentially accessible regions

The identification of differentially accessible regions (DARs) between the conditions was done using DiffBind v.2.16.2 (*Stark, 2011*) and edgeR v.3.30.1 (*Robinson et al., 2010*). Log fold changes were calculated, and their associated p-values were corrected for multiple hypothesis testing via the Benjamini–Hochberg procedure to obtain adjusted p-values. The DARs were annotated by their nearest gene using the *annotatePeaks.pl* function of Homer v.4.11 (*Heinz et al., 2010*).

## Gene set enrichment analysis

Genes nearby the DARs were ranked based on the log-fold change calculated with edgeR v.3.30.1 (*Robinson et al., 2010*). This ranked list of genes was used as input to perform gene set enrichment analysis with the *fgseaMultilevel* function of the R package fgsea v.1.14.0 (*Korotkevich et al., 2021*). The *FGSEA-multilevel* method is based on an adaptive multi-level split Monte Carlo scheme, which allows the estimation of very low p-values. The Hallmark gene sets collection from the Molecular Signatures Database (MSigDB) (*Korotkevich et al., 2021*) was used as a reference to identify the biological processes that were significantly enriched.

## Motif enrichment analysis

The identification of known TF motifs found in peaks overlapping the promoter region (±5 kb of TSS) was done using the *findMotifsGenome.pl* function from HOMER v.4.9.1 (*Heinz et al., 2010*). The *-size* parameter was set to *given* to use the exact peak region as target sequence. Following the screening of HOMER's reliable motifs library against the target sequences, the motifs enriched with a p-value less than 0.05 are returned.

## Animal care

All procedures that were performed on animals were approved by the McGill University Animal Care Committee (UACC), protocol #7512.

## Acknowledgements

We thank Christian Young at the Lady Davis Institute for Medical Research—Jewish General Hospital-core facility for his help with fluorescence-activated cell sorting (FACS) of muscle stem cells. We thank Dr. Michael Witcher at McGill University Department of Oncology for his critical comments and careful review of an early draft of this manuscript.

## Additional information

### Funding

| Funder | Grant reference number | Author |
|---|---|---|
| Natural Sciences and Engineering Research Council of Canada | | Vahab D Soleimani |

The funders had no role in study design, data collection and interpretation, or the decision to submit the work for publication.

### Author contributions

Korin Sahinyan, Darren M Blackburn, Formal analysis, Investigation, Methodology, Visualization, Writing - original draft, Writing – review and editing; Marie-Michelle Simon, Data curation, Formal analysis, Investigation, Software, Visualization, Writing – review and editing; Felicia Lazure, Writing – review and editing; Tony Kwan, Investigation, Validation; Guillaume Bourque, Supervision; Vahab D Soleimani, Conceptualization, Formal analysis, Funding acquisition, Investigation, Methodology, Project administration, Resources, Supervision, Validation, Writing – review and editing

### Author ORCIDs

Korin Sahinyan http://orcid.org/0000-0002-2641-5250
Darren M Blackburn http://orcid.org/0000-0001-9084-7595
Guillaume Bourque http://orcid.org/0000-0002-3933-9656
Vahab D Soleimani http://orcid.org/0000-0003-2154-4894

### Ethics

All procedures that were performed on animals were approved by the McGill University Animal Care Committee (UACC), protocol #7512.

### Decision letter and Author response

Decision letter https://doi.org/10.7554/eLife.72792.sa1
Author response https://doi.org/10.7554/eLife.72792.sa2

# Additional files

### Supplementary files

• Transparent reporting form

• Source data 1. Quality control source data. (A) Unlabeled agarose gel (1.25%) of MuSC ATAC-Seq sequence ready libraries. (B) Unlabeled agarose gel (1.25%) of uninjured myofiber ATAC-Seq sequence ready library. (C) Labeled agarose gel (1.25%) image of MuSC and uninjured myofiber ATAC-Seq sequence ready libraries. (D) Raw file of bioanalyzer results from single myofiber sequence ready ATAC-Seq libraries.

### Data availability

The data discussed in this study have been deposited in NCBI's Gene Expression Omnibus and are accessible through GEO Series accession numbers GSE173676 and GSE171534.

The following datasets were generated:

| Author(s) | Year | Dataset title | Dataset URL | Database and Identifier |
|---|---|---|---|---|
| Soleimani V | 2021 | ATAC-Seq of single myofibers | https://www.ncbi.nlm.nih.gov/geo/query/acc.cgi?acc=GSE173676 | NCBI Gene Expression Omnibus, GSE173676 |
| Soleimani V | 2021 | ATAC-Seq of young and aged satellite cells | https://www.ncbi.nlm.nih.gov/geo/query/acc.cgi?acc=GSE171534 | NCBI Gene Expression Omnibus, GSE171534 |

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
