## [Editor Report]

The authors have described an innovative application of ATAC-Seq for genome-wide analysis of the chromatin state at single myofiber resolution.

---

## [Decision Letter]

**Decision letter after peer review:**

Thank you for submitting your article "Application of ATAC-Seq for genome-wide analysis of the chromatin state at single myofiber resolution" for consideration by *eLife*. Your article has been reviewed by 3 peer reviewers, and the evaluation has been overseen by a Reviewing Editor and Y M Dennis Lo as the Senior Editor. The following individual involved in review of your submission has agreed to reveal their identity: Nora Yucel (Reviewer #3).

The reviewers have discussed their reviews with one another, and the Reviewing Editor has drafted this to help you prepare a revised submission. The majority opinion of the group is that the article would potentially be suitable as a Tools and Resources article in *eLife*.

Essential revisions:

1) The authors state multiple times that this technique gives good sequencing depth. As such, information should be provided regarding number of high quality reads per sample, and whether replicates were downsampled before peak calling.

2) In Figure 1—figure supplement 1A, the x-axis label is not showing the correct size of the library.

3) Can the authors provide the smfATAC-Seq genome tracks for Myod1 and Myog loci to illustrate the chromatin accessibility changes in uninjured and injured myofibers since these two genes are essential for muscle regeneration?

4) In Line 169, remove redundant "can be" in the sentence.

5) The gene names in Figure 5C are not shown clearly.

6) The authors should consider a deeper analysis of the differential chromatin accessibility peaks (subdivided as promoters and distal regions), including prediction of TFs binding sites and integration with other appropriate datasets exploring epigenetic mechanisms (such as histone marks). In addition, the differential ATAC-seq peaks (mainly the ones overlapping putative promoters) should be combined with similar datasets exploring transcriptional changes and used to better infer gene networks characterizing the experimental groups. For example, by generating smfATAC-seq data from a slow-twitching muscle (Soleus), they could take advantage of available transcriptomic (https://journals.plos.org/plosone/article?id=10.1371/journal.pone.0016807) and proteomic (https://www.embopress.org/doi/full/10.15252/embr.201439757) datasets.

7) Chromatin accessibility analysis in satellite cells was performed by isolating cells from the mouse hindlimb. However, these data were compared to the smfTAC-seq from EDL myofibers. The authors should acknowledge the limitation of this comparison.

8) If the authors intend for smfATACSeq to be performed broadly, it might be helpful to put it on a resource like https://www.protocols.io/ for other researchers to easily use and add notes.

9) It would be valuable to systematically compare enrichment at non-myogenic promoters to emphasize the myonuclear enrichment. This could be shown by overlaying TSS enrichment plots for genes that are characteristic of myogenic, vs immune vs vascular cells.

10) Total skeletal muscle ATAC-Seq has been previously published (Ramachandran, et al., PLoS Biology, 2019) in a variety of skeletal muscle types (including EDL vs soleus). How does smfATAC data compare to this ATACSeq data-- in particular are EDL-specific peaks also observed?

11) The authors should include motif analysis of differential chromatin regions (uninjured vs injured, mdx vs WT, using unchanged regions as background). The authors state (line 341) that "Despite the increase in chromatin accessibility during injury, the accessible chromatin regions in both injured and uninjured fibers are associated with genes involved in similar biological processes." Motif analyses may in this case more be informative than GO, and could identify transcription factors with differential activity in the various experimental conditions.

*Reviewer #1 (Recommendations for the authors):*

1) The authors claim that the smfATAC-seq provides a high sequencing depth that allows for peak calling and differential peak analysis. Can the authors provide the exact sequencing depth in the method section?

2) In Figure 1—figure supplement 1A, the x-axis label is not showing the correct size of the library.

3) The concordance of the smfATAC-Seq data does not seem very good. In Figure 3, only 2 replicates are provided for uninjured fibers and they seem entirely separate. In Figure 6D, WT fibers also do not cluster very well. The authors should provide more replicates for the same condition to show the reproducibility of the approach.

4) In Figure 3F, there are very few unique peaks in uninjured fibers compared with MuSCs. Does it mean the uninjured fibers are less accessible than MuSCs, or is it due to the different nuclei number input for ATAC-seq? This comparison is not simple and the authors should use the same nuclei number input for different samples.

5) Can the authors provide the smfATAC-Seq genome tracks for Myod1 and Myog loci to illustrate the chromatin accessibility changes in uninjured and injured myofibers since these two genes are essential for muscle regeneration?

6) In MDX mice, the myofibers undergo regeneration and degeneration cycles. Currently, the authors only compare the uninjured WT myofibers and the MDX myofibers. Can the authors also provide a detailed comparison between the CTX injured myofibers and the MDX myofibers to illustrate the difference in the chromatin accessibility profiles of the two conditions, if any? Alternatively, when is the peak of regeneration and degeneration cycles during the life of an MDX mouse? Again, the current study lacks depth and shows no biological insight.

7) In Line 169, remove redundant "can be" in the sentence.

8) The gene names in Figure 5C are not shown clearly.

*Reviewer #2 (Recommendations for the authors):*

The authors should consider a deeper analysis of the differential chromatin accessibility peaks (subdivided as promoters and distal regions), including prediction of TFs binding sites and integration with other appropriate datasets exploring epigenetic mechanisms (such as histone marks).

In addition, the differential ATAC-seq peaks (mainly the ones overlapping putative promoters) should be combined with similar datasets exploring transcriptional changes and used to better infer gene networks characterizing the experimental groups. For example, by generating smfATAC-seq data from a slow-twitching muscle (Soleus), they could take advantage of available transcriptomic (https://journals.plos.org/plosone/article?id=10.1371/journal.pone.0016807) and proteomic (https://www.embopress.org/doi/full/10.15252/embr.201439757) datasets.

*Reviewer #3 (Recommendations for the authors):*

The authors have done a good job in demonstrating data quality, and making it available through GEO. The methods are also quite clear. If the authors intend for smfATACSeq to be performed broadly, it might be helpful to put it on a resource like https://www.protocols.io/ for other researchers to easily use and add notes. Other specific points in no particular order:

– In Figure 3B PCA is done on MuSCs vs uninjured vs injured fibers. It looks like the MuSCs are driving the differences in PC1, perhaps compressing the differences in fibers. Does injury segregate samples by PCA when the MuSCs are removed? On a similar note, in figure 6D, if injured fibers are included in the PCA along with mdx fibers (normalizing for what I assume are different sequencing preparations), are they intermediate to the mdx fibers, as stated in lines 384-386?

– It would be valuable to systematically compare enrichment at non-myogenic promoters to emphasize the myonuclear enrichment. This could be shown by overlaying TSS enrichment plots for genes that are characteristic of myogenic, vs immune vs vascular cells.

– Total skeletal muscle ATAC-Seq has been previously published (Ramachandran, et al., PLoS Biology, 2019) in a variety of skeletal muscle types (including EDL vs soleus). How does smfATAC data compare to this ATACSeq data-- in particular are EDL-specific peaks also observed?

– The authors should include motif analysis of differential chromatin regions (uninjured vs injured, mdx vs WT, using unchanged regions as background). The authors state (line 341) that "Despite the increase in chromatin accessibility during injury, the accessible chromatin regions in both injured and uninjured fibers are associated with genes involved in similar biological processes." Motif analyses may in this case more be informative than GO, and could identify transcription factors with differential activity in the various experimental conditions.

– The authors state multiple times that this technique gives good sequencing depth. As such, information should be provided regarding number of high quality reads per sample, and whether replicates were downsampled before peak calling.

---

## [Author Response]

Essential revisions:1) The authors state multiple times that this technique gives good sequencing depth. As such, information should be provided regarding number of high quality reads per sample, and whether replicates were downsampled before peak calling.

The detailed sequencing specifications were provided in Table 1. However, we have edited the text to refer to the number of reads when the sequencing depth is mentioned.

For instance, the text on the lines 86-87 reads as:

“The smfATAC-Seq has a sequencing depth of approximately 6 million final reads aligned that provides approximately 30 000 peaks called”

As well as the text on the lines 109-111:

“Following the removal of mitochondrial reads, there were approximately 6 million final reads aligned and 30 000 peaks called, demonstrating a sufficient sequencing depth for downstream analysis (Table 1).”

2) In Figure 1—figure supplement 1A, the x-axis label is not showing the correct size of the library.

We thank the reviewer for their careful observation. The labelling for the 300 bp was not placed correctly on the x-axis during the editing process. We have now placed the labels in the correct positions on the x-axis. The correct library size can now be accurately visualized on Figure 1—figure supplement 1A where the bioanalyzer peaks correspond to approximately 200 bp, 400 bp and 600 bp representing regions with mono-, di- and tri-nucleosomes respectively.

3) Can the authors provide the smfATAC-Seq genome tracks for Myod1 and Myog loci to illustrate the chromatin accessibility changes in uninjured and injured myofibers since these two genes are essential for muscle regeneration?

We have now included IGV tracks for all the Myogenic Regulatory Factors (MRFs) in Figure 2 —figure supplement 3. We have included these results in the text on lines 211-217 which reads as follows:

“Muscle regeneration and repair rely on the temporal expression of Myogenic Regulatory factors (MRFs), Myf5, MyoD, Myog and Myf6/MRF4 (1-3). Therefore, we assessed the chromatin accessibility of the MRFs in MuSCs and in the myofibers under homeostasis and regeneration (Figure 2—figure supplement 3). We observed peaks in the promoter regions of Myf5 only in the MuSCs but not in the myofibers and peaks in the promoters of Myog and Myf6/MRF4 were solely observed in the myofibers (Figure 2—figure supplement 3). However, we observed peaks in the promoter regions of MyoD in both the MuSCs and myofibers (Figure 2—figure supplement 3).”

4) In Line 169, remove redundant "can be" in the sentence.

We have now removed the redundant wording in the revised text, please see line 242.

5) The gene names in Figure 5C are not shown clearly.

We have now edited the figure to make the gene names clearer and readable.

6) The authors should consider a deeper analysis of the differential chromatin accessibility peaks (subdivided as promoters and distal regions), including prediction of TFs binding sites and integration with other appropriate datasets exploring epigenetic mechanisms (such as histone marks). In addition, the differential ATAC-seq peaks (mainly the ones overlapping putative promoters) should be combined with similar datasets exploring transcriptional changes and used to better infer gene networks characterizing the experimental groups. For example, by generating smfATAC-seq data from a slow-twitching muscle (Soleus), they could take advantage of available transcriptomic (https://journals.plos.org/plosone/article?id=10.1371/journal.pone.0016807) and proteomic (https://www.embopress.org/doi/full/10.15252/embr.201439757) datasets.

We have now performed comparative analysis between our smfATAC-Seq and ChIPSeq performed on EDL muscle for H3K27ac by Ramachandran, et al., 2019. The figure for this analysis can be found in Figure 2 —figure supplement 2K. We now discuss the results of this analysis in the lines 184- 191 which reads as follows:

“Accessible chromatin regions are associated with various histone marks such as H3K27ac and H3K4me3 (4-6). Thus, we compared the smfATAC-Seq to publicly available datasets on ChIP-Seq on H3K27ac in EDL muscle that was previously performed by Ramachandran, et al., 2019 (GSM3515022, GSM3515023) (7). The comparative analysis has revealed that there were only 97 peaks in the smfATAC-Seq that did not overlap with the H3K27ac peaks, while the majority of the peaks, 6090 peaks, were common to the H3K27ac peaks present in the entire EDL muscle (Figure 2—figure supplement 2K). This demonstrates that the accessible regions that are assessed by smfATAC-Seq correspond to the regions of the chromatin marked by histones that are associated with open chromatin such as H3K27ac.”

7) Chromatin accessibility analysis in satellite cells was performed by isolating cells from the mouse hindlimb. However, these data were compared to the smfTAC-seq from EDL myofibers. The authors should acknowledge the limitation of this comparison.

We do understand the reviewers comment that MuSCs and myofibers are different. However, since muscle fibers are derived from fusion of muscle stem cells during development and regeneration, we find it interesting to compare the chromatin states between these different cell types.

Since our manuscript is a “tools and resource” article, our main aim in inclusion of the MuSCs ATAC-Seq data was to compare the quality of the conventional OMNI ATAC-Seq on 5000 cells to our new protocol on a single myofiber that contains only about 200-300 myonuclei. Through this comparison, we show that smfATAC-Seq can also provide high resolution assessment of chromatin accessibility a single muscle fiber

8) If the authors intend for smfATACSeq to be performed broadly, it might be helpful to put it on a resource like https://www.protocols.io/ for other researchers to easily use and add notes.

We agree with the reviewer, and we are preparing a step-by-step protocol to be submitted to Bio-protocol.

9) It would be valuable to systematically compare enrichment at non-myogenic promoters to emphasize the myonuclear enrichment. This could be shown by overlaying TSS enrichment plots for genes that are characteristic of myogenic, vs immune vs vascular cells.

In addition to the figure that we presented in the initial submission (Figure 2—figure supplement 1), we have now looked at the enrichment of non- myogenic genes more globally. First, we compared the enrichment of ATAC-Seq peaks on non-myogenic genes in our smfATAC-Seq with an independent ATAC-Seq performed on whole EDL muscle by Ramachandran et al., 2019 (GSM3981673) (7) (Table 2) which contains non myogenic muscle resident cells in addition to the myogenic cells as it is representative of the whole muscle. The results of this comparison can be seen in Table 2.

We have discussed the results of this analysis in the lines 127-143 which reads as follows:

“Given that the whole muscle contains non-myogenic cell types, we first compared smfATAC-Seq to an ATAC-Seq performed on whole EDL muscle from Ramachandran, et al., 2019 (GSM3981673) (7) for the enrichment of ATAC-Seq peaks on the genes of non-myogenic cells. We obtained the list of genes that are solely expressed in the whole muscle (RPM of at least 10) but not in the myofibers (RPM of 0) by using an RNA-Seq dataset performed on whole muscle and a single myofiber by Blackburn, et al., 2019 (GSE138591) (8). This list represents genes that are only expressed by the muscle resident non-myogenic cell types, designated as non-fiber muscle genes. We determined the number of peaks in smfATAC-Seq that overlap with non-fiber muscle genes which revealed that only 0.1% of the peaks overlapped with the top 100 non-fiber muscle genes (Table 2). In comparison, 0.33% of the peaks in the whole EDL muscle ATAC-Seq

(GSM3981673) (7) overlapped with the top 100 non-fiber muscle genes (Table 2). The significant difference in the overlap with the non-fiber muscle genes between the whole muscle ATAC-Seq and smfATAC-Seq suggest that the whole muscle ATAC-Seq has enrichment of peaks associated with non-myogenic genes when compared to the smfATAC-Seq which implies that smfATAC-Seq can successfully exclude the non-myogenic cell types. In contrast, number of overlapping peaks with all the genes expressed in whole muscle in the EDL ATAC-Seq and smfATAC-Seq were similar (Table 2).”

10) Total skeletal muscle ATAC-Seq has been previously published (Ramachandran, et al., PLoS Biology, 2019) in a variety of skeletal muscle types (including EDL vs soleus). How does smfATAC data compare to this ATACSeq data-- in particular are EDL-specific peaks also observed?

We have now compared our smfATAC-Seq on the uninjured myofiber to the whole EDL muscle ATAC-Seq that was mentioned by the reviewer (Ramachandran, et al., 2019). We have determined that 65% of the peaks in the smfATAC-Seq overlap with the ATAC-Seq of the whole muscle by at least 1 bp. We have included a detailed table on the overlap between the two datasets (Table 3).

We discuss the results of this comparison in the lines 179-182:

“We also analyzed the overlap between the smfATAC-Seq on single EDL myofibers with the ATAC-Seq performed on the whole EDL muscle by Ramachandran, et al., 2019 (GSM3981673) (7). This analysis revealed that 65% of the smfATAC-Seq peaks in the single uninjured myofibers overlap with the whole EDL muscle ATAC-Seq (Table 3).”

11) The authors should include motif analysis of differential chromatin regions (uninjured vs injured, mdx vs WT, using unchanged regions as background). The authors state (line 341) that "Despite the increase in chromatin accessibility during injury, the accessible chromatin regions in both injured and uninjured fibers are associated with genes involved in similar biological processes." Motif analyses may in this case more be informative than GO, and could identify transcription factors with differential activity in the various experimental conditions.

We have now performed motif analysis and presented the new data in new Figure 4 —figure supplement 4. This data is included in the text in the lines 298-306 which reads as follows:

“Furthermore, we analyzed the enrichment of transcription factor binding motifs in the sequences under peaks common between the injured and uninjured myofibers overlapping the promoters (+/- 5kb of TSS) (Figure 4—figure supplement 2A) as well as in the peaks that are unique to injured and uninjured myofibers overlapping the promoters (+/- 5kb of TSS) (Figure 4—figure supplement 2B). The top motifs that were enriched in the sequences under peaks common to injured and uninjured myofibers include binding site for Mef2a (Figure 4—figure supplement 2A). On the other hand, the top motifs that were enriched in the sequences under peaks unique to injured myofibers included binding sites for JUN and Stat3 (Figure 4—figure supplement 2B). However, due to the low number of unique peaks in the uninjured myofibers

(Figure 3F), there was no significant motif that enriched in that peak set.”

We have also performed motif analysis for WT and MDX myofibers in Figure 6 —figure supplement 5, in lines 384-390:

“Finally, we determined the top motifs that are enriched in the sequences under the peaks that are common between the mdx and WT myofibers overlapping the promoters (+/- 5 kb of TSS) (Figure 6—figure supplement 5A) as well as the sequences under peaks that are unique to mdx and WT overlapping the promoters (+/- 5 kb of TSS) (Figure 6—figure supplement 5B-C). The top significantly enriched motifs in the peaks common between mdx and WT included Mef2a and JUN (Figure 6—figure supplement 5A) while the top motifs enriched in the peaks unique to mdx included transcription factors such as Foxo1 (Figure 6—figure supplement 5B).”

Reviewer #1 (Recommendations for the authors):1) The authors claim that the smfATAC-seq provides a high sequencing depth that allows for peak calling and differential peak analysis. Can the authors provide the exact sequencing depth in the method section?

The detailed sequencing specifications were provided in Table 1. However, we have edited the text to refer to the number of reads when the sequencing depth is mentioned.

For instance, the text on the lines 86-87 reads as: “The smfATAC-Seq has a sequencing depth of approximately 6 million final reads aligned that provides approximately 30 000 peaks called” as well as the text on the lines 109-111:

“Following the removal of mitochondrial reads, there were approximately 6 million final reads aligned and 30 000 peaks called, demonstrating a sufficient sequencing depth for downstream analysis (Table 1).”

2) In Figure 1—figure supplement 1A, the x-axis label is not showing the correct size of the library.

We thank the reviewer for their careful observation. The labelling for the 300 bp was not placed correctly on the x-axis during the editing process. We have now placed the labels in the correct positions on the x-axis. The correct library size can now be accurately visualized on Figure 1—figure supplement 1A where the bioanalyzer peaks correspond to approximately 200 bp, 400 bp and 600 bp representing regions with mono-, di- and tri-nucleosomes respectively.

3) The concordance of the smfATAC-Seq data does not seem very good. In Figure 3, only 2 replicates are provided for uninjured fibers and they seem entirely separate. In Figure 6D, WT fibers also do not cluster very well. The authors should provide more replicates for the same condition to show the reproducibility of the approach.

It is well established that there is a high degree of variation between the myofibers of skeletal muscles. A recent study of RNA-seq on a single myofiber exhibited variations between the individual myofibers within the same condition (8). In addition, another recent study using snRNA-seq found great variation in the transcriptome of myonuclei from the same myofibers (9). In our smfATAC-Seq, while the two uninjured fibers are not identical, they are still very similar as they vary only along the principal component 2 which represents a variation of only 10%. Similarly, while the WT myofibers in Figure 6D also display heterogeneity, they are still more closely associated to the other WT myofibers than to the mdx myofibers.

4) In Figure 3F, there are very few unique peaks in uninjured fibers compared with MuSCs. Does it mean the uninjured fibers are less accessible than MuSCs, or is it due to the different nuclei number input for ATAC-seq? This comparison is not simple and the authors should use the same nuclei number input for different samples.

The difference in the number of unique peaks between uninjured myofibers and MuSCs are possibly due to the difference in the number of input nuclei. OMNI ATAC-Seq was performed on 5000 MuSCs while the smfATAC-Seq was performed on a single myofiber that contains 200-300 myonuclei. The reason for the inclusion of the MuSC ATAC-Seq data in this manuscript was for the purpose of quality control comparison.

5) Can the authors provide the smfATAC-Seq genome tracks for Myod1 and Myog loci to illustrate the chromatin accessibility changes in uninjured and injured myofibers since these two genes are essential for muscle regeneration?

We have now included IGV tracks for all the Myogenic Regulatory Factors (MRFs) in Figure 2 —figure supplement 3. We have included these results in the text on lines 211-217 which reads as follows:

“Muscle regeneration and repair rely on the temporal expression of Myogenic Regulatory factors (MRFs), Myf5, MyoD, Myog and Myf6/MRF4 (1-3). Therefore, we assessed the chromatin accessibility of the MRFs in MuSCs and in the myofibers under homeostasis and regeneration (Figure 2—figure supplement 3). We observed peaks in the promoter regions of Myf5 only in the MuSCs but not in the myofibers and peaks in the promoters of Myog and Myf6/MRF4 were solely observed in the myofibers (Figure 2—figure supplement 3). However, we observed peaks in the promoter regions of MyoD in both the MuSCs and myofibers (Figure 2—figure supplement 3).”

6) In MDX mice, the myofibers undergo regeneration and degeneration cycles. Currently, the authors only compare the uninjured WT myofibers and the MDX myofibers. Can the authors also provide a detailed comparison between the CTX injured myofibers and the MDX myofibers to illustrate the difference in the chromatin accessibility profiles of the two conditions, if any? Alternatively, when is the peak of regeneration and degeneration cycles during the life of an MDX mouse? Again, the current study lacks depth and shows no biological insight.

The focus of a “Tools and Resource” paper is to validate a new method and not necessarily provide novel biological insights or discoveries. Due to the difference in the genetic background of the mice of injured (C57/BL6) and mdx (C57BL/10ScSn-Dmdmdx/J), the comparison between them might not be appropriate and informative. However, since GO term analysis revealed similar processes between injured and mdx myofibers, we assessed the overall difference in chromatin accessibility by performing heatmap clustering of Pearson correlation coefficients and the PCA analysis on mdx, WT and injured myofibers (Figure 6 —figure supplement 4). Figure 6 —figure supplement 4B shows that while mdx and WT myofibers are more similar to each other than they are to the injured myofibers. However the injured myofibers were more similar to mdx than they are to the WT myofibers. In the future, it would be interesting to perform smfATAC-Seq on mdx and WT mice that are injured in order to compare the myofiber chromatin accessibility during regeneration in a diseased context.

7) In Line 169, remove redundant "can be" in the sentence.

We have now removed the redundant wording in the revised text, please see line 242.

8) The gene names in Figure 5C are not shown clearly.

We have now edited the figure to make the gene names clearer and readable.

Reviewer #2 (Recommendations for the authors):The authors should consider a deeper analysis of the differential chromatin accessibility peaks (subdivided as promoters and distal regions), including prediction of TFs binding sites and integration with other appropriate datasets exploring epigenetic mechanisms (such as histone marks).In addition, the differential ATAC-seq peaks (mainly the ones overlapping putative promoters) should be combined with similar datasets exploring transcriptional changes and used to better infer gene networks characterizing the experimental groups. For example, by generating smfATAC-seq data from a slow-twitching muscle (Soleus), they could take advantage of available transcriptomic (https://journals.plos.org/plosone/article?id=10.1371/journal.pone.0016807) and proteomic (https://www.embopress.org/doi/full/10.15252/embr.201439757) datasets.

We have now performed comparative analysis between our smfATAC-Seq and ChIPSeq performed on EDL muscle for H3K27ac by Ramachandran, et al., 2019. The figure for this analysis can be found in Figure 2 —figure supplement 2K. We now discuss the results of this analysis in the lines 184-191 which reads as follows:

“Accessible chromatin regions are associated with various histone marks such as H3K27ac and H3K4me3 (4-6). Thus, we compared the smfATAC-Seq to publicly available datasets on ChIP-Seq on H3K27ac in EDL muscle that was previously performed by Ramachandran, et al., 2019 (GSM3515022, GSM3515023) (7). The comparative analysis has revealed that there were only 97 peaks in the smfATAC-Seq that did not overlap with the H3K27ac peaks, while the majority of the peaks, 6090 peaks, were common to the H3K27ac peaks present in the entire EDL muscle (Figure 2—figure supplement 2K). This demonstrates that the accessible regions that are assessed by smfATAC-Seq correspond to the regions of the chromatin marked by histones that are associated with open chromatin such as H3K27ac.”

Reviewer #3 (Recommendations for the authors):The authors have done a good job in demonstrating data quality, and making it available through GEO. The methods are also quite clear. If the authors intend for smfATACSeq to be performed broadly, it might be helpful to put it on a resource like https://www.protocols.io/ for other researchers to easily use and add notes.

We agree with the reviewer, and we are preparing a manuscript detailing step-by-step protocol to be submitted to Bio-protocol.

Other specific points in no particular order:– In Figure 3B PCA is done on MuSCs vs uninjured vs injured fibers. It looks like the MuSCs are driving the differences in PC1, perhaps compressing the differences in fibers. Does injury segregate samples by PCA when the MuSCs are removed? On a similar note, in figure 6D, if injured fibers are included in the PCA along with mdx fibers (normalizing for what I assume are different sequencing preparations), are they intermediate to the mdx fibers, as stated in lines 384-386?

We have now performed the PCA analysis solely with uninjured and injured myofibers, without the MuSC samples (Figure 3 —figure supplement 1). The injured and uninjured myofibers cluster separately, showing their overall difference.

Findings of this analysis is discussed in the lines 225-229 which reads as follows:

“To test whether the differences between regenerating and resting myofibers are overshadowed by their differences with MuSCs, we performed heatmap clustering of Pearson correlation coefficients and PCA analysis for injured and uninjured myofibers only, without MuSCs (Figure 3—figure supplement 1). This further highlighted how the uninjured and injured myofibers cluster separately (Figure 3—figure supplement 1).”

Furthermore, we have also performed the PCA analysis with mdx, WT and injured myofibers in new Figure 6 —figure supplement 4. Findings of this analysis is discussed in the lines 374-382 which reads as:

“Since the observed differential biological processes between WT and mdx myofibers were similar to those seen between injured vs uninjured myofibers, we then compared the overall differences in chromatin accessibility between mdx, WT, and injured myofibers. We performed heatmap clustering of Pearson correlation and PCA analysis between WT, mdx and injured myofibers (Figure 6—figure supplement 4). These analyses have revealed that injured myofibers were more similar to the mdx than they are to the WT C57BL/10ScSn myofibers. However, as expected due to the different backgrounds of the mice between injured and the WT and mdx mice, WT and mdx myofibers were more similar to each other than they are to the injured C57BL/6 myofibers (Figure 6—figure supplement 4).”

– It would be valuable to systematically compare enrichment at non-myogenic promoters to emphasize the myonuclear enrichment. This could be shown by overlaying TSS enrichment plots for genes that are characteristic of myogenic, vs immune vs vascular cells.

In addition to the figure that we presented in the initial submission (Figure 2—figure supplement 1), we have now looked at the enrichment of non-myogenic genes more globally. First, we compared the enrichment of ATAC-Seq peaks on non-myogenic genes in our smfATAC-Seq with an independent ATAC-Seq performed on whole EDL muscle by Ramachandran et, al. 2019 (GSM3981673) (7) (Table 2) which contains non myogenic muscle resident cells in addition to the myogenic cells as it is representative of the whole muscle. The results of this comparison can be seen in Table 2.

We have discussed the results of this analysis in the lines 127-143 which reads as follows:

“Given that the whole muscle contains non-myogenic cell types, we first compared smfATACSeq to an ATAC-Seq performed on whole EDL muscle from Ramachandran, et al., 2019 (GSM3981673) (7) for the enrichment of ATAC-Seq peaks on the genes of non-myogenic cells. We obtained the list of genes that are solely expressed in the whole muscle (RPM of at least 10) but not in the myofibers (RPM of 0) by using an RNA-Seq dataset performed on whole muscle and a single myofiber by Blackburn, et al., 2019 (GSE138591) (8). This list represents genes that are only expressed by the muscle resident non-myogenic cell types, designated as non-fiber muscle genes. We determined the number of peaks in smfATAC-Seq that overlap with non-fiber muscle genes which revealed that only 0.1% of the peaks overlapped with the top 100 non-fiber muscle genes (Table 2). In comparison, 0.33% of the peaks in the whole EDL muscle ATAC-Seq

(GSM3981673) (7) overlapped with the top 100 non-fiber muscle genes (Table 2). The significant difference in the overlap with the non-fiber muscle genes between the whole muscle ATAC-Seq and smfATAC-Seq suggest that the whole muscle ATAC-Seq has enrichment of peaks associated with non-myogenic genes when compared to the smfATAC-Seq which implies that smfATAC-Seq can successfully exclude the non-myogenic cell types. In contrast, number of overlapping peaks with all the genes expressed in whole muscle in the EDL ATAC-Seq and smfATAC-Seq were similar (Table2).”

– Total skeletal muscle ATAC-Seq has been previously published (Ramachandran, et al., PLoS Biology, 2019) in a variety of skeletal muscle types (including EDL vs soleus). How does smfATAC data compare to this ATACSeq data-- in particular are EDL-specific peaks also observed?

We have now compared our smfATAC-Seq on the uninjured myofiber to the whole EDL muscle ATAC-Seq that was mentioned by the reviewer (Ramachandran, et al., 2019). We have determined that 65% of the peaks in the smfATAC-Seq overlap with the ATAC-Seq of the whole muscle by at least 1 bp. We have included a detailed table on the overlap between the two datasets (Table 3).

We discuss the results of this comparison in the lines 179-182:

“We also analyzed the overlap between the smfATAC-Seq on single EDL myofibers with the ATAC-Seq performed on the whole EDL muscle by Ramachandran, et al., 2019 (GSM3981673) (7). This analysis revealed that 65% of the smfATAC-Seq peaks in the uninjured myofiber overlap with the whole EDL muscle ATAC-Seq (Table 3).”

– The authors should include motif analysis of differential chromatin regions (uninjured vs injured, mdx vs WT, using unchanged regions as background). The authors state (line 341) that "Despite the increase in chromatin accessibility during injury, the accessible chromatin regions in both injured and uninjured fibers are associated with genes involved in similar biological processes." Motif analyses may in this case more be informative than GO, and could identify transcription factors with differential activity in the various experimental conditions.

We have now performed motif analysis and presented the new data in new Figure 4 —figure supplement 4. This data is included in the text in the lines 298-306 which reads as follows:

“Furthermore, we analyzed the enrichment of transcription factor binding motifs in the sequences under peaks common between the injured and uninjured myofibers overlapping the promoters (+/- 5kb of TSS) (Figure 4—figure supplement 2A) as well as in the peaks that are unique to injured and uninjured myofibers overlapping the promoters (+/- 5kb of TSS) (Figure 4—figure supplement 2B). The top motifs that were enriched in the sequences under peaks common to injured and uninjured myofibers include binding site for Mef2a (Figure 4—figure supplement 2A). On the other hand, the top motifs that were enriched in the sequences under peaks unique to injured myofibers included binding sites for JUN and Stat3 (Figure 4—figure supplement 4B). However, due to the low number of unique peaks in the uninjured myofibers

(Figure 3F), there was no significant motif that enriched in that peak set.”

We have also performed motif analysis for WT and mdx myofibers in Figure 6 —figure supplement 5, in lines 384-390:

“Finally, we determined the top motifs that are enriched in the sequences under the peaks that are common between the mdx and WT myofibers overlapping the promoters (+/- 5 kb of TSS) (Figure 6—figure supplement 5A) as well as the sequences under peaks that are unique to mdx and WT overlapping the promoters (+/- 5 kb of TSS) (Figure 6—figure supplement 5B-C). The top significantly enriched motifs in the peaks common between mdx and WT included Mef2a and JUN (Figure 6—figure supplement 5A) while the top motifs enriched in the peaks unique to mdx included transcription factors such as Foxo1 (Figure 6—figure supplement 5B).”

– The authors state multiple times that this technique gives good sequencing depth. As such, information should be provided regarding number of high quality reads per sample, and whether replicates were downsampled before peak calling.

The detailed sequencing specifications were provided in Table 1. However, we have edited the text to refer to the number of reads when the sequencing depth is mentioned.

For instance, the text on the lines 86-87 reads as:

“The smfATAC-Seq has a sequencing depth of approximately 6 million final reads aligned that provides approximately 30 000 peaks called” as well as the text on the lines 109-111: “Following the removal of mitochondrial reads, there were approximately 6 million final reads aligned and 30 000 peaks called, demonstrating a sufficient sequencing depth for downstream analysis (Table 1).”

References:

1. Hernandez-Hernandez, J. M., Garcia-Gonzalez, E. G., Brun, C. E., and Rudnicki, M. A. (2017) The myogenic regulatory factors, determinants of muscle development, cell identity and regeneration. *Semin Cell Dev Biol* 72, 10-18

2. Montarras, D., Chelly, J., Bober, E., Arnold, H., Ott, M. O., Gros, F., and Pinset, C. (1991) Developmental patterns in the expression of Myf5, MyoD, myogenin, and MRF4 during myogenesis. *New Biol* 3, 592-600

3. Asfour, H. A., Allouh, M. Z., and Said, R. S. (2018) Myogenic regulatory factors: The orchestrators of myogenesis after 30 years of discovery. *Exp Biol Med (Maywood)* 243, 118-128

4. Zhang, T., Cooper, S., and Brockdorff, N. (2015) The interplay of histone modifications – writers that read. *EMBO Rep* 16, 1467-1481

5. Berger, S. L. (2007) The complex language of chromatin regulation during transcription. *Nature* 447, 407-412

6. Barrera, L. O., Li, Z., Smith, A. D., Arden, K. C., Cavenee, W. K., Zhang, M. Q., Green, R. D., and Ren, B. (2008) Genome-wide mapping and analysis of active promoters in mouse embryonic stem cells and adult organs. *Genome Res* 18, 46-59

7. Ramachandran, K., Senagolage, M. D., Sommars, M. A., Futtner, C. R., Omura, Y., Allred, A. L., and Barish, G. D. (2019) Dynamic enhancers control skeletal muscle identity and reprogramming. *PLoS Biol* 17, e3000467

8. Blackburn, D. M., Lazure, F., Corchado, A. H., Perkins, T. J., Najafabadi, H. S., and Soleimani, V. D. (2019) High-resolution genome-wide expression analysis of single myofibers using SMART-Seq. *J Biol Chem* 294, 20097-20108

9. Petrany, M. J., Swoboda, C. O., Sun, C., Chetal, K., Chen, X., Weirauch, M. T., Salomonis, N., and Millay, D. P. (2020) Single-nucleus RNA-seq identifies transcriptional heterogeneity in multinucleated skeletal myofibers. *Nat Commun* 11, 6374